# Oligomeric-solvent engineering of hierarchical hydrogen-bonding networks for multifunctional glass interlayers

Min Li, Longyu Hu, Menghan Pi, Xiayue Yang, Xiaoyu He, Wei Cui ✉ & Rong Ran ✉

The properties of polymer gels are governed not only by the crosslinked network but also by the solvent. Conventional small-molecule solvents impose trade-offs among environmental adaptability, cost, and biocompatibility. Here, we employ oligomeric polyethylene glycol as a multifunctional solvent for poly (methacrylic acid) (PMAA), converting the otherwise plastic PMAA network into a transparent, dissipative gel. The oligomeric solvent promotes a hierarchical hydrogen-bonding architecture with broadly distributed strengths, coupling elasticity and viscosity, stabilizing the network, and enabling high energy dissipation for acoustic damping and impact resistance. Meanwhile, thermally reversible hydrogen-bond dissociation provides broad endothermic heat absorption, affording thermal buffering. The gels further exhibit high transparency, robust adhesion, and self-healing. Harnessing the gel as an interlayer, we fabricate laminated glass that integrates light transmission, thermal regulation, sound attenuation, and mechanical protection. This oligomer-solvent strategy offers a practical route to multifunctional, energy-efficient, safer building glazing applications in real-world architecture settings.

Polymer gels are soft materials composed of cross-linked polymer networks swollen with a solvent phase[1]. The polymer network forms the structural backbone, while the solvent occupies the interstitial spaces[2]. Importantly, the solvent does more than simply fill space. It plays a pivotal role in determining the physicochemical properties, functional performance, and practical applications of the gel. Specifically, (1) solvent molecules infiltrate the polymer network, counteracting elastic retraction of the crosslinks through swelling and thereby preserving macroscopic volume and shape[3]; (2) solvents critically influence responsiveness of the gel to environmental stimuli (e.g. temperature, pH, light, electric fields) by mediating molecular interactions and phase transitions[4]; (3) solvents modulate mechanical properties through polymer-solvent interactions[5–7]; and (4) solvents can serve as functional carriers, enabling drug delivery[8], ionic conduction[9], and functioning as media for in situ catalysis or material synthesis[10,11].

Common gel solvents are typically small molecules such as water or organic liquids[12], each with characteristic drawbacks. For instance, hydrogels are flexible and biocompatible but suffer from water crystallization at low temperature and evaporation at high temperature, which narrows the operating window, as well as thermodynamic incompatibility with hydrophobic components[13,14]. Organogels can mitigate drying and compatibility issues but often exhibit weak polymer-solvent interactions and limited biocompatibility[15]. Ionic liquids and deep eutectic solvents offer ultralow volatility and useful functionalities, yet they present practical drawbacks including cost, synthetic complexity, biodegradability concerns, composition sensitivity, and high viscosity[16–20]. These trade-offs motivate solvent

College of Polymer Science and Engineering, State Key Laboratory of Polymer Materials Engineering, Sichuan University, Chengdu, China.
✉e-mail: cuiwei@scu.edu.cn; ranrong@scu.edu.cn

strategies that better balance performance, processability, environmental adaptability, and biocompatibility.

Motivated by the inherent limitations of conventional small-molecule solvents, recent efforts have increasingly turned to polymeric liquids as alternative gel solvents. Liu et al.[21] introduced poly(n-butyl acrylate) (PBA) fluid into polymer networks, achieving efficient energy dissipation over a broad frequency range. Wang et al.[22–24] established PEG as a polymeric solvent platform for gel systems, demonstrating that PEG-based gels can simultaneously exhibit high mechanical strength, self-healing capability, energy dissipation, and 3D printability. Although these studies highlight the versatility of polymeric solvents, their design targets and underlying mechanisms differ from those of the present work. In PEG gels based on P(HEMA-co-AAc)[23], enhanced toughness and self-healing primarily originate from solvent-enabled multivalent interactions and increased chain correlations or entanglements. In contrast, polymer-fluid-based damping gels[21] achieve tunable dissipation by regulating the relaxation (reptation) dynamics of confined polymeric liquids, leading to molecular-weight-dependent and frequency-shiftable dissipation peaks.

Here, we pursue a distinct molecular design strategy: we select PMAA, whose carboxylic groups intrinsically form strong associative assemblies[25] (bulk copolymerization of MAA yields glassy, brittle PMAA, Fig. 1a), and employ PEG-200 as an oligomeric solvent that introduces additional weaker/medium-strength hydrogen-bonding states, thereby broadening the bond-strength distribution within a single interaction family (hydrogen bonding). First, owing to its oligomeric nature, PEG has far greater contour length than small-molecule solvents and more effectively separates adjacent PMAA chains, thereby increasing interchain spacing and free volume[26,27]. Second, the ether oxygen on PEG bears lone pairs and serves as hydrogen-bonding acceptors, forming multiple bidentate (medium-strength) and monodentate (weak) interactions with the carboxylic groups of PMAA[28]. In parallel, a fraction of PMAA carboxyl groups associate with each other from cyclic dimers to linear aggregates[25,29], supplying the strong-bond tier of the hierarchy (Fig. 1b). Notably, both PEG and PMAA are indispensable for building this hierarchical hydrogen-bonding network. Substituting PEG with $H_2O$ results in a glassy, brittle PMAA material rather than a gel (Supplementary Fig. 1), while replacing PMAA with polyacrylic acid (PAAc) suppresses gelation in the PEG/PAAc (PEA) system (Supplementary Fig. 2). This architecture achieves a finely balanced elasticity and viscosity[30,31], enabling a gel-like response across an ultra-wide frequency range (Fig. 1c). The elastic component provides high mechanical strength, while the viscous component delivers high damping, with a loss factor (tan δ) approaching 1. This combination confers both noise reduction and impact energy absorption. Simultaneously, the thermally responsive interactions of the PEG/PMAA

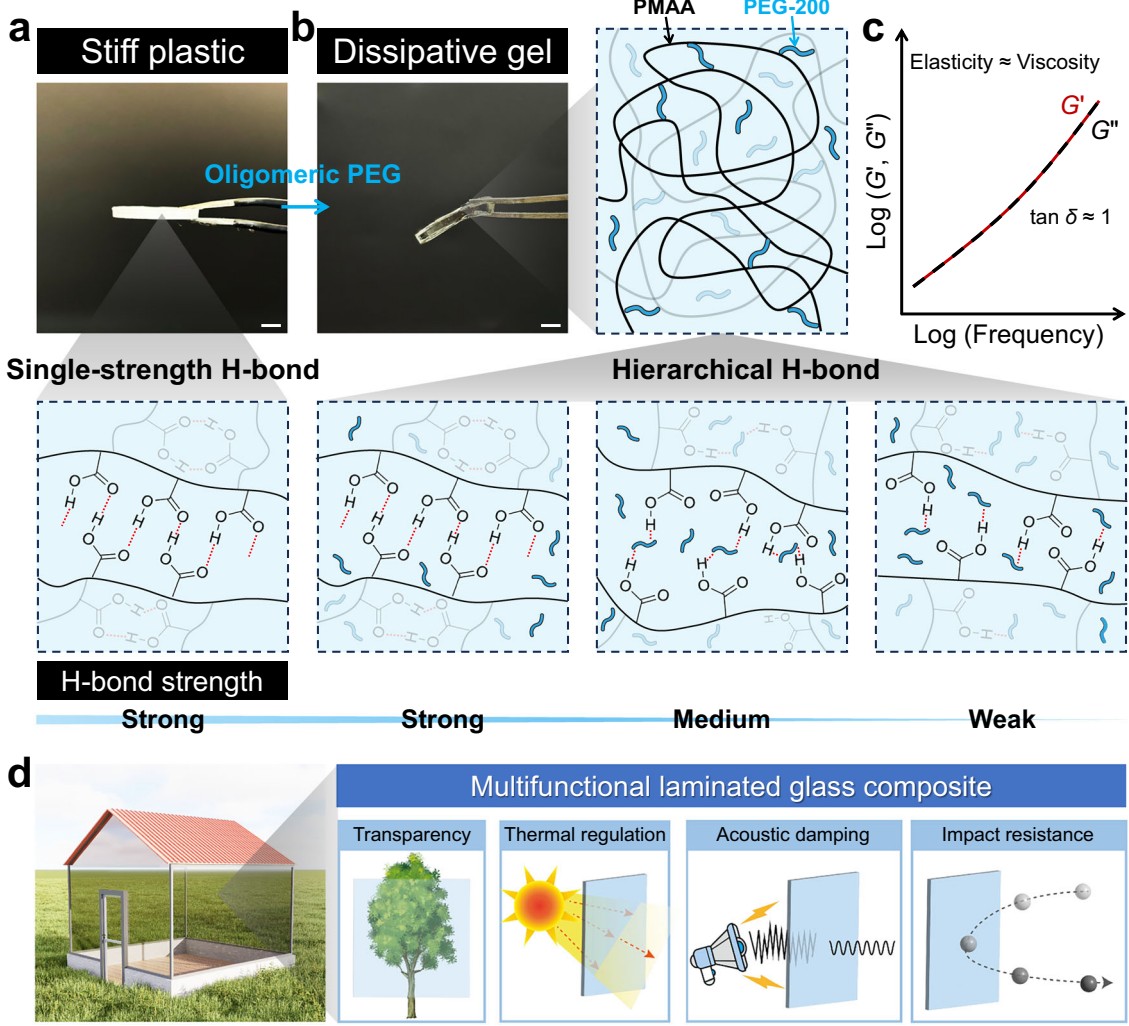

**Fig. 1 | Design of a hierarchical hydrogen-bond network in PMAA. a** Photograph and schematic of a pure PMAA network featuring a single-strength hydrogen-bonding network. **b** Photograph and schematic of a PMAA-based hierarchical hydrogen-bonding network containing oligomeric PEG as the solvent. **c** Schematic illustration of frequency-independent damping behavior enabled by hierarchical hydrogen bonding. **d** Schematic of multifunctional glass with integrated properties of high transparency, thermal regulation, acoustic damping, and impact resistance. Scale bars in photographs represent 1 mm.

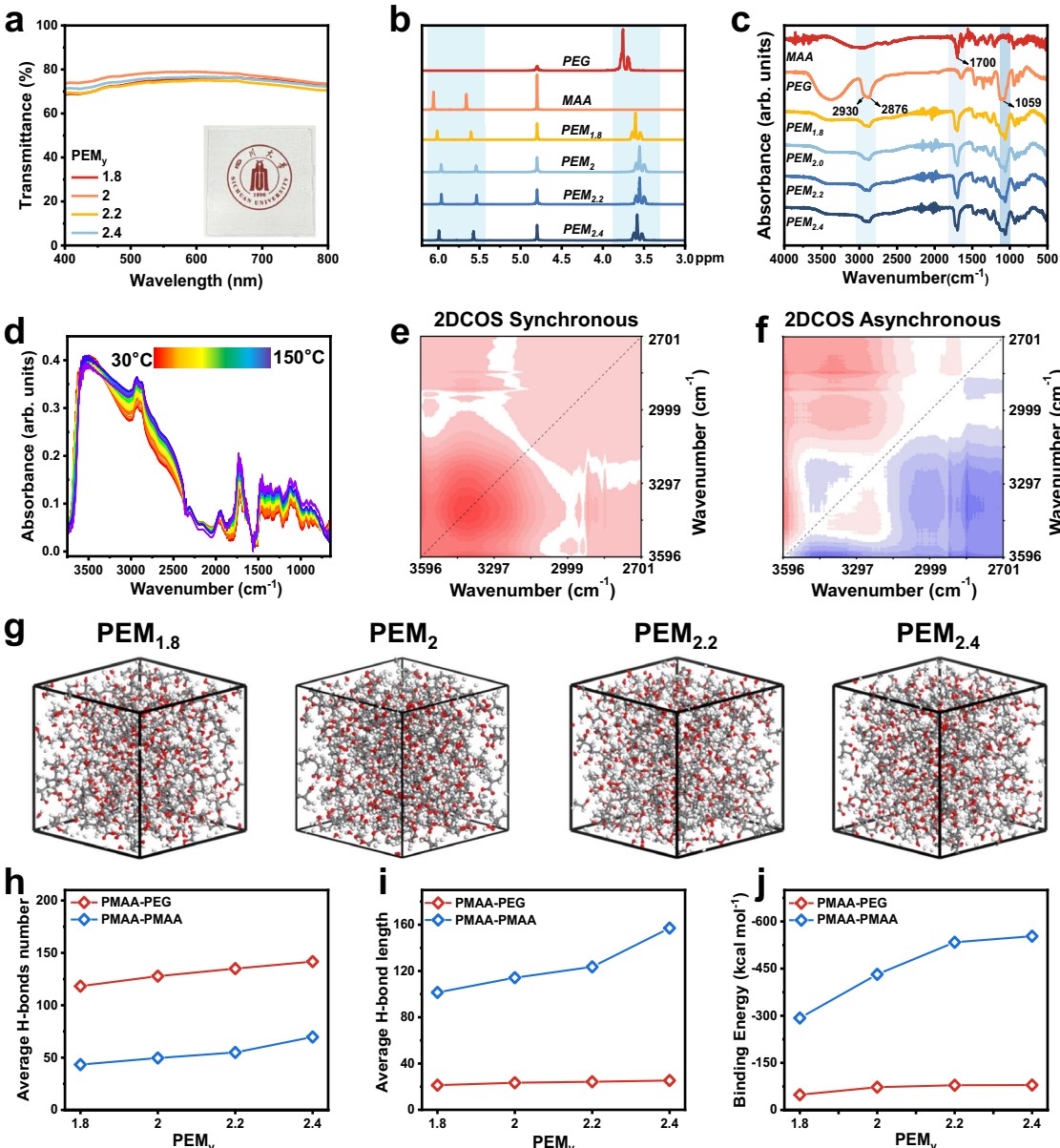

**Fig. 2 | Characterizations and simulations revealing hierarchical hydrogen-bonding in PEM_y gels. a** UV-vis transmittance of all PEM_y gels in the wavelength range of 400-800 nm. Inset: a logo clearly visible beneath a PEM₂ gel film (2 mm thick). **b** ¹H NMR spectra of representative samples. **c** FTIR spectra of the gel components and PEM_y gels. **d** Temperature-dependent FTIR spectra of PEM₂ gel upon heating from 30 to 150 ºC (10 ºC intervals). **e, f** Synchronous and asynchronous 2DCOS spectra of the PEM₂ gel. Red regions indicate positive intensities, while blue regions represent negative ones. **g** Molecular dynamics (MD) simulations of PEM_y gels. Atom colors: H, white; C, gray; O, red. **h, i** Average hydrogen-bond number and average hydrogen-bond length between PMAA–PEG and PMAA–PMAA. **j** Binding energy between PMAA–PEG, and PMAA–PMAA.

system enhance thermal regulation. To demonstrate practical utility, we integrated the gel into a laminated-glass prototype and systematically evaluated its multifunctionality (Fig. 1d). The resulting composite exhibits a combination of optical clarity, thermal buffering, acoustic damping, and mechanical robustness. Rather than optimizing isolated properties, this system achieves integrated performance, opening avenues for functional materials in architectural glazing, protective interfaces, and energy-responsive systems.

## Results

### Hierarchical hydrogen-bonding network: design and characterization

We characterized the structure of PEM_y gels across different length scales. Optical transmittance in the 400–800 nm range was measured

using an ultraviolet-visible spectrophotometer. All PEM_y gels exhibit high transparency, with a logo image clearly visible beneath a 2 mm-thick PEM₂ gel film (Fig. 2a). Successful gel formation was further verified by ¹H nuclear magnetic resonance (NMR) spectroscopy (Fig. 2b). The multiple peaks near 3.70 ppm correspond to protons in PEG[32], while the peaks at 6.06 ppm and 5.66 ppm arise from MAA. In the PEM_y gels, all three signals remain but shift downfield, confirming network formation and their participation in hydrogen bond interactions. Fourier transform infrared (FT-IR) spectroscopy further elucidates the chemical composition and interaction mechanism within the PEM_y gels (Fig. 2c). For neat PEG, absorption peaks at 2876 cm⁻¹ and 2930 cm⁻¹ correspond to symmetrical and asymmetrical stretching of −CH₂ groups, while the peak at 1059 cm⁻¹ is assigned to C–O–C stretching[33,34]. For neat MAA, the characteristic C = O stretching

vibration appears near 1700 cm⁻¹. These characteristic peaks are present in all PEM$_y$ gel samples, confirming the successful incorporation of both components.

To clarify the hierarchical nature of the hydrogen-bonding network, peak deconvolution was performed on the C=O vibration FT-IR spectra. Neat MAA exhibits a prominent carbonyl peak at 1700 cm⁻¹ (Supplementary Fig. 3a), characteristic of strong dimeric hydrogen bonds. Upon PEG incorporation, an additional blue-shifted peak emerges near 1728 cm⁻¹ (Supplementary Fig. 3b-e). This shift arises from the disruption of partial strong PMAA–PMAA dimers and the formation of weaker PEG–PMAA hydrogen bonds, which diminish the electron-withdrawing environment of the carbonyl group[35]. Peak-area ratio analysis further reveals that the fraction of strongly hydrogen-bonding carbonyl groups increases with MAA content (Supplementary Fig. 3f), underscoring the compositional dependence of hydrogen-bonding interactions within the gel network. Concomitantly, the C–O–C stretching of PEG (1000–1100 cm⁻¹) undergoes pronounced broadening and becomes increasingly asymmetric with rising MAA content, accompanied by the appearance of a shoulder at lower wavenumbers (Supplementary Fig. 4)[36,37]. This feature signifies the coexistence of free ether oxygen and hydrogen-bonding ether oxygen, reflecting hydrogen bond interactions between PEG ether oxygens and PMAA hydroxyl groups.

To elucidate the thermal evolution of this hierarchical hydrogen-bonding network at the molecular level, temperature-dependent FT-IR spectra of the PEM$_2$ gel were recorded from 30 to 150 °C. Focusing first on the O–H stretching region of PMAA (Fig. 2d), a gradual blue shift from 3500 to 3515 cm⁻¹ indicates the progressive dissociation of hydrogen-bond aggregates into free carboxyl groups. Two-dimensional correlation spectroscopy (2DCOS) provided higher resolution of these sequential events. While synchronous spectra (Fig. 2e) indicate cooperative spectral changes across the broad 2700–3600 cm⁻¹ range, the asynchronous spectra (Fig. 2f) distinguish sequential thermal events: the spectral response of hydrogen bond O–H groups (-3297 cm⁻¹) precedes that of the weakly associated or free O–H groups (-3596 cm⁻¹), corroborating the progressive thermal disruption of the network.

Further insight is provided by the C=O stretching region (1690–1750 cm⁻¹, Supplementary Fig. 5). The synchronous spectra exhibit positive cross-peaks between the band at -1700 cm⁻¹ (dimeric PMAA–PMAA hydrogen bonds) and the band at -1728–1730 cm⁻¹ (free C=O and PEG–PMAA hydrogen bonds), indicating concurrent intensity variations during heating. According to Noda's rule, the positive synchronous and positive asynchronous correlations at (1700, 1728 cm⁻¹) reveal the specific reaction sequence: the destabilization of hydrogen-bonding PMAA–PMAA carbonyls initiates first, followed by the subsequent development of free or PEG–PMAA carbonyls.

Then, we further investigated hydrogen-bonding interactions between PMAA and PEG chains through molecular dynamics (MD) simulations. Representative simulation snapshots are presented in Fig. 2g. The evolution of hydrogen-bonding numbers over a 100 ps trajectory is shown in Supplementary Fig. 6, and the corresponding average hydrogen-bonding counts are summarized in Fig. 2h. On the one hand, the number of hydrogen bonds between PMAA and PEG is consistently greater than that between PMAA chains, indicating that the introduction of PEG disrupts the dense interchain hydrogen bonds in PMAA and inserts itself as a bridging component. On the other hand, both types of hydrogen bonds increase with higher MAA content, reflecting the growing availability of hydrogen-bonding donors and acceptors within the network. Figure 2i presents the average hydrogen-bonding lengths. The PMAA–PMAA hydrogen bonds exhibit longer average distances than those between PMAA–PEG, further supporting the notion that PEG effectively inserts between PMAA chains and establishes shorter, more favorable hydrogen bonds as molecular bridges. To complement these findings, binding energies

between PMAA–PEG and PMAA–PMAA were calculated via molecular simulation, as shown in Fig. 2j. The binding energy increases with the number of hydrogen bonds in both cases. Notably, the binding energy of PMAA–PMAA is significantly higher than that of PMAA–PEG. This difference is attributed to the formation of strong, multidentate hydrogen-bonding assemblies. Combined with the bond length analysis, these results confirm the hierarchical nature of hydrogen bonding within the PEG–PMAA network.

Furthermore, scanning electron microscopy (SEM) images reveal that the cross sections of PEM$_y$ gels are flat and smooth, with no observable signs of phase separation (Supplementary Fig. 7). The morphological uniformity is further supported by small-angle X-ray scattering (SAXS) results of the PEM$_2$ gel (Supplementary Fig. 8). Thermalgravimetric analysis (TGA) demonstrates that PEM$_y$ gels possess good thermal stability (Supplementary Fig. 9). The glass transition temperature ($T_g$) of PEM$_y$ gels increases with increasing MAA content and remains close to room temperature (Supplementary Fig. 10). This trend is attributed to the increasing density of hydrogen bonds within the network. Since viscoelastic material exhibits maximum internal energy dissipation near their $T_g$, these gels are promising candidates for room-temperature damping applications.

It is worth noting that hierarchical interactions have also been reported in several recent gel systems[38–40], such as ion–dipole hybrid gels and deep eutectic solvent–based gels. In these materials, hierarchy is typically introduced through chemical heterogeneity, for example by combining multiple interaction motifs (ionic coordination, ion–dipole coupling, or eutectic complexation) with distinct bonding energies. By contrast, we adopt a fundamentally different molecular design strategy in which hierarchical behavior emerges from a single interaction type—hydrogen bonding—whose strength distribution is continuously broadened through oligomeric-solvent engineering. The finite chain length of oligomeric PEG molecules enables them to function simultaneously as a solvent, spacer, and multidentate hydrogen-bonding bridge, generating hierarchy through topological and length-scale modulation. This strategy provides a conceptually simple and general route to hierarchical viscoelastic networks.

## Mechanical properties, flaw insensitivity and self-healing

We next investigated the mechanical properties, flaw-insensitivity and self-healing of PEM$_y$ gels. The mechanical performance was first analyzed through uniaxial tensile testing, and the stress-strain curves for all PEM$_y$ gels are summarized in Fig. 3a. With increasing MAA content, the fracture stress of PEM$_y$ gel increases and the yielding behavior becomes more pronounced. Under an applied stress field, network failure proceeds from low to high binding energy. Therefore, the hydrogen bonds within the gel network rupture first, dissipating energy and weakening interchain interactions, which leads to yield initiation. Then the external force is transmitted along the long-entangled chains until the covalent bonds at the crosslink points are broken, causing the disentanglement and interchain slippage. Eventually, excessive covalent bonds are broken to form voids in the network, causing the gel to break. More PMAA chains result in an increased number of hydrogen bonds and chain entanglements, which together enhance both the Young's modulus and toughness of the PEM$_y$ gels (Supplementary Fig. 11). To further analyze the stress-strain behavior, we applied the Mooney–Rivlin equation (Fig. 3b), expressed as follows[41,42]:

$$\sigma_{red} = \frac{\sigma}{\lambda - \lambda^{-2}} = 2C_1 + 2C_2\frac{1}{\lambda} \tag{1}$$

where $\sigma_{red}$ is the reduced stress, $\lambda$ is the stretch ratio, $C_1$ is the material constant corresponding to half of the shear modulus, and $C_2$ reflects strain hardening ($C_2 < 0$) or softening ($C_2 > 0$) beyond the Gaussian elasticity regime[43]. As shown in Supplementary Fig. 12, the

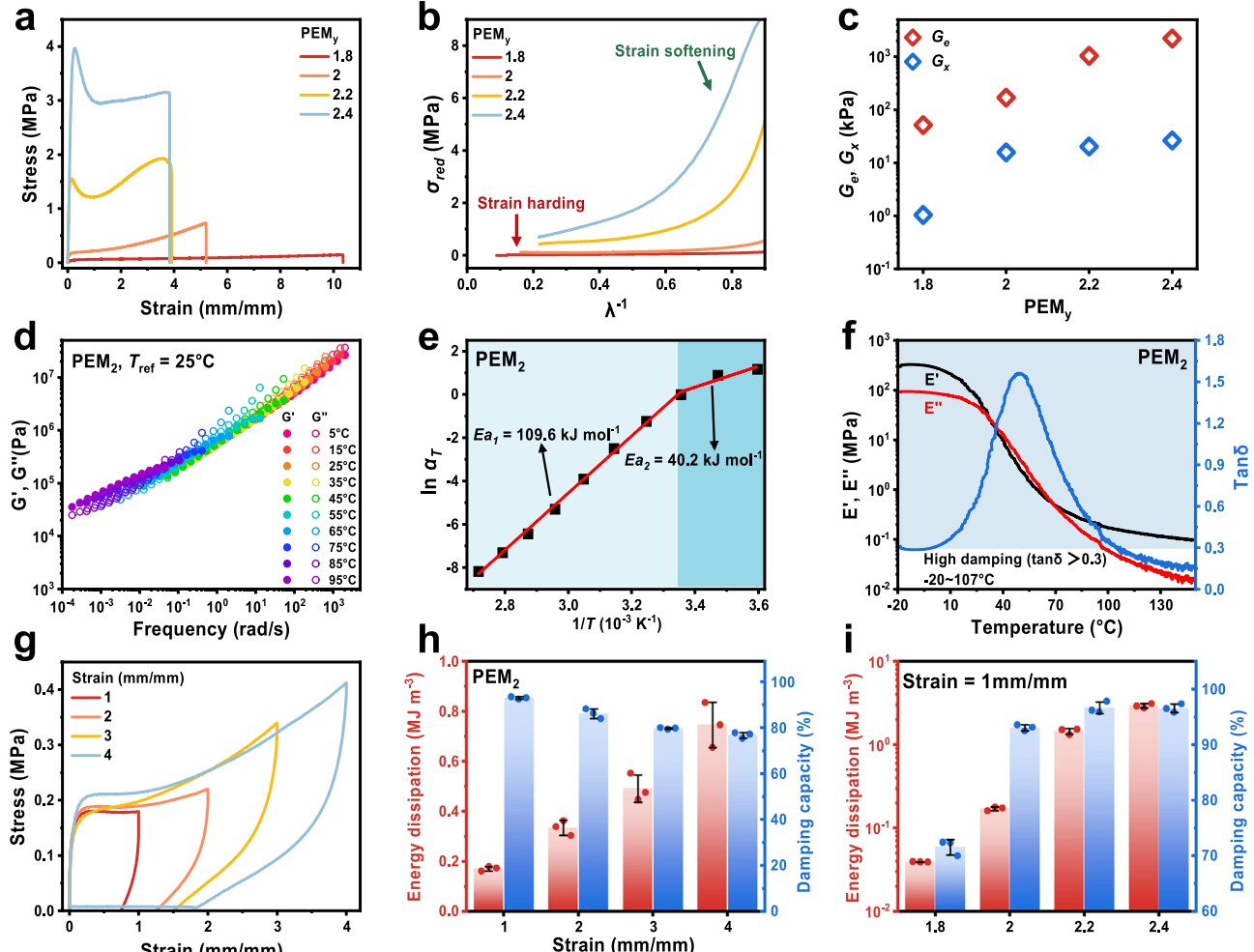

**Fig. 3 | Mechanical properties and viscoelastic behavior of PEM$_y$ gels. a** Tensile stress-strain curves of PEM$_y$ gel samples. **b** Mooney-Rivlin plots of PEM$_y$ gels. **c** $G_e$ and $G_x$ values of PEM$_y$ gels obtained by fitting with the Rubinstein-Panyukov model. **d** Rheological master curve of the PEM$_2$ gel constructed using the time-temperature superposition principle at a reference temperature of 25 °C. **e** Fitting curve for the apparent activation energy of the PEM$_2$ gel. **f** Temperature-dependent DMA result of the PEM$_2$ gel between -20-150 °C. **g** Tensile loading-unloading cycle curves of the PEM$_2$ gel at different fixed strains. **h** Summary of energy dissipation values and damping capacity of the PEM$_2$ gel with different tensile strains. **i** Summary of energy dissipation values and damping capacity of all PEM$_y$ gels at strain of 1 mm/mm. Data are presented as the mean values ± SD, n = 3 independent samples. Source data are provided as a Source Data file.

Mooney−Rivlin plots reveal that PEM$_{2.2}$ and PEM$_{2.4}$ exhibit strain softening across the entire strain range, PEM$_2$ gel shows softening only at high 1/$\lambda$ and PEM$_{1.8}$ shows neither strain softening nor hardening. For gels with higher MAA content (particularly PEM$_{2.2}$ and PEM$_{2.4}$), strain softening arises from the rupture of interchain hydrogen bonds. In contrast, as the MAA content decreases, the number of hydrogen bonds decreases and chain motion becomes increasingly restricted by chemical crosslinks, making strain softening insignificant (PEM$_2$) or even absent (PEM$_{1.8}$). Because only a small amount of chemical crosslinker was used, the permanent crosslink density is low, positioning PEM$_{1.8}$ at the boundary between strain softening and hardening ($C_2 \approx 0$).

To gain deeper insight into the gel network structure, the Rubinstein−Panyukov model was also used to analyze the stress-strain curves. The model is expressed as[44]:

$$\frac{\sigma}{\lambda - \lambda^{-2}} = G_x + \frac{G_e}{0.74\lambda + 0.61\lambda^{-0.5} - 0.35} \quad (2)$$

where $\sigma$ is the stress, $\lambda$ is the stretch ratio, and $G_x$ and $G_e$ represent the shear modulus contributions from crosslinks and entanglements,

respectively[45]. The fitted curves are shown in Supplementary Fig. 13, and the extracted values of $G_x$ and $G_e$ are summarized in Fig. 3c. Clearly, $G_e$ is consistently larger than $G_x$, and both increase with higher MAA content. This indicates that entanglements dominate the mechanical properties of PEM$_y$ gels, while the limited number of chemical crosslinks primarily serve to stabilize the network. With increasing PMAA content, hydrogen bonding interactions become more abundant, and the resulting hydrogen-bonding aggregates act in a manner similar to covalent crosslinks. These aggregates not only function as additional crosslinking sites but also promote entanglement by bandaging chains together, thereby suppressing disentanglement during deformation.

The densely packed hydrogen bonds and extensive chain entanglements not only enhance the mechanical strength of the PEM$_y$ gel but also promote crack passivation. To evaluate the defect tolerance of the gel, a single-edge notch test was conducted using the PEM$_2$ gel as a representative example (Supplementary Fig. 14a). The notched gel exhibited a fracture strain of 3.56 mm/mm, approximately 70% of that of the unnotched sample, demonstrating that the PEM$_2$ gel is relatively insensitive to structural flaws. The corresponding fracture energy was calculated to be 3.52 kJ m$^{-2}$. In addition, the notched gel was able to lift

a 200 g weight without visible crack propagation (Supplementary Fig. 14b), further confirming its flaw-insensitive behavior.

The PEM$_y$ gel also exhibits excellent self-healing capability at room temperature, enabled by the dynamic and reversible nature of its hydrogen bonds. Notably, no external stimuli such as heat or solvents are required to trigger healing. The healing process was monitored in situ using optical microscopy (Supplementary Fig. 15a). Initially, the PEM$_2$ gel was completely severed into two parts, and the width of the crack gradually narrowed over time. After 24 h, the crack was almost fully closed. The mechanical recovery was quantified by uniaxial tensile tests (Supplementary Fig. 15b), and the self-healing efficiency was calculated based on the recovery of fracture strain (Supplementary Fig. 15c). The PEM$_2$ gel reached a self-healing efficiency of 84% after 24 h at room temperature.

## Viscoelastic behavior and energy dissipation

The viscoelastic behavior of PEM$_y$ gels was investigated. To determine the relaxation time, stress relaxation tests were conducted under a fixed strain of 1 mm/mm. As shown in Supplementary Fig. 16, the relaxation time decreases with increasing MAA content, suggesting that dissociation of hydrogen bonds plays a dominant role in stress relaxation. Rheological measurements were performed to characterize the viscoelastic nature of the gels. The shear modulus, defined as the sum of the storage modulus ($G'$) and the loss modulus ($G''$), reflects the material's combined elasticity and viscous responses. In most cases, when $G'$ exceeds $G''$, the material behaves as a predominantly elastic solid. Conversely, when $G''$ exceeds $G'$, viscous behavior dominates, indicating a viscous (sol) state[46]. As shown in Fig. 3d and Supplementary Fig. 17, all PEM$_y$ gels exhibit a near-constant gel-like state across an ultra-wide frequency range from $10^{-4}$ to $10^3$ Hz, spanning eight orders of magnitude. Within this range, $G'$ and $G''$ remain comparable, and the loss factor (tan $\delta$) fluctuates around unity (Supplementary Fig. 18). This indicates that PEM$_y$ gels possess a well-balanced viscoelastic profile: they are capable of dissipating mechanical energy while also maintaining structural recovery without flowing freely[30,31]. A relevant mechanistic interpretation is provided in Supplementary Fig. 19.

Furthermore, the apparent activation energy of all PEM$_y$ gels was calculated (Fig. 3e and Supplementary Fig. 20). Each gel exhibited two distinct activation energies, with the high-temperature region showing larger values than the low-temperature region. This behavior can be ascribed to multiple relaxation mechanisms[47]. At high temperatures (corresponding to low frequencies), hydrogen bonds and covalent crosslinks gradually dissociate, and the activation energy ($Ea_1$) primarily reflects bond dissociation. These values are significantly higher than the typical dimerization energy of -COOH groups (approximately 25-40 kJ mol$^{-1}$)[48]. In contrast, at low temperatures (corresponding to high frequencies), relaxation is dominated by chain entanglements, and the activation energy ($Ea_2$) corresponds to the relaxation of long polymer chains.

To gain deeper insight into the damping performance of the PEM$_y$ gel, dynamic mechanical analysis (DMA) was performed. As shown in Fig. 3f and Supplementary Fig. 21 a-c, both the storage modulus ($E'$) and loss modulus ($E''$) decrease markedly with increasing temperature, in agreement with the time–temperature equivalence principle and the rheological observations. The loss factor (tan $\delta$) presents a clear peak near the $T_g$ for all PEM$_y$ gels, and regions with tan $\delta > 0.3$ are defined as high-damping regions. All PEM$_y$ gels demonstrate good damping performance across a broad temperature range. Similar to the DSC results, increasing the MAA content shifts $T_g$ to higher temperatures (Supplementary Fig. 21d), reflecting the enhanced strength and density of hydrogen-bonding interactions. The viscoelastic nature of the PEM$_2$ gel was further verified by varying both the tensile rate and the testing temperature (Supplementary Fig. 22). The mechanical response exhibited strong dependencies on both variables: the gel behaved in a hard and brittle manner under high tensile rates and low temperatures,

and became soft and tough under low tensile rates and high temperatures. These trends are fully consistent with the time-temperature equivalence principle.

As an intuitive manifestation of the damping capability, energy dissipation serves as a key criterion for evaluating the damping performance of materials. In tensile loading–unloading cycles, the hysteresis loops enlarge with strain (Fig. 3g and Supplementary Fig. 23a-c). The damping capacity, defined as the ratio of dissipated energy to total strain energy, decreases with increasing strain (Fig. 3h and Supplementary Fig. 23d-e). Two factors contribute to this result. First, at small strain, the polymer chains remain coiled, enabling substantial internal friction and interchain sliding, which leads to pronounced energy dissipation and high damping capacity. With increasing strain, chains gradually elongate and orient, reducing available sliding space and thereby lowering frictional dissipation. Second, at small strain, the dynamic and reversible crosslinks within the gel can continuously break and reform, contributing significantly to energy dissipation. With increasing strain, many of these weak interactions are rapidly ruptured and cannot readily reform, thereby diminishing the dissipation capacity. Meanwhile, the network is increasingly supported by the polymer backbone and strong crosslinks, causing the gel network to exhibit more elasticity. Notably, we observed at the same strain, the damping capacity increases with MAA content (Fig. 3i), underscoring the key role of hydrogen bond density in enhancing energy dissipation.

Consistent with tensile observations, large-amplitude oscillatory shear (LAOS) tests further demonstrate the dissipation capability of the PEM$_y$ gels (Supplementary Fig. 24). All gels exhibit stable elliptical Lissajous loops over 100 cycles, indicating stable linear viscoelastic behavior[49]. The loop area, corresponding to the dissipated energy per cycle ($\Delta W$), decreases only slightly during cycling, confirming that the reversible hydrogen bonds repeatedly break and reform without structural damage. Moreover, $\Delta W$ increases with MAA content, demonstrating that a higher density of hydrogen-bonding interactions enhances energy dissipation.

Furthermore, we emphasize that our oligomeric-solvent engineering strategy is universal. To examine its generality, PEGs with different molecular weights (400, 600, and 800 g mol$^{-1}$) were employed to prepare PE$_x$M$_y$ gels. Longer PEG chains act as plasticizers, increasing the distance between PMAA chains and thereby reducing the number of hydrogen bonds formed. As a result, replacing PEG-200 with PEG-400, PEG-600, and PEG-800 in the PEM$_2$ formulation yielded sol-like rather than gel-like products (Supplementary Fig. 25). To obtain comparable gel states, the MAA content was accordingly increased, resulting in the PE$_{400}$M$_3$, PE$_{600}$M$_4$, and PE$_{800}$M$_5$ formulations. All of these gels display high transparency (Supplementary Fig. 26a). Their stress-strain curves confirm robust, gel-like toughness (Supplementary Fig. 26b), and they exhibit excellent damping capacity (Supplementary Fig. 26c). The rheological behavior of PE$_{400}$M$_3$, PE$_{600}$M$_4$, and PE$_{800}$M$_5$ is summarized in Supplementary Fig. 27, showing that all maintain a balance between viscosity and elasticity across an ultra-wide frequency range, with tan $\delta$ values approaching 1.

## Thermal management, impact protection, and acoustic absorption properties

The hierarchical hydrogen-bonding network in PE$_x$M$_y$ gels gradually dissociates with increasing temperature, resulting in an overall endothermic response. This feature suggests that PE$_x$M$_y$ gels hold promise for thermal management applications. To verify their endothermic capability, disc-shaped PE$_x$M$_y$ gel samples (diameter = 2 cm) were placed on a white substrate and exposed to a xenon lamp for 10 min (1000 W m$^{-2}$). As shown in Fig. 4a, thermal infrared images reveal maximum surface temperatures of 35.9 °C, 34.8 °C, 33.1 °C, and 31.8 °C, respectively, despite the ambient temperature exceeding 80 °C. This confirms the effective heat-buffering ability of the gels. A broad endothermic peak was also observed in the DSC curves of all PE$_x$M$_y$

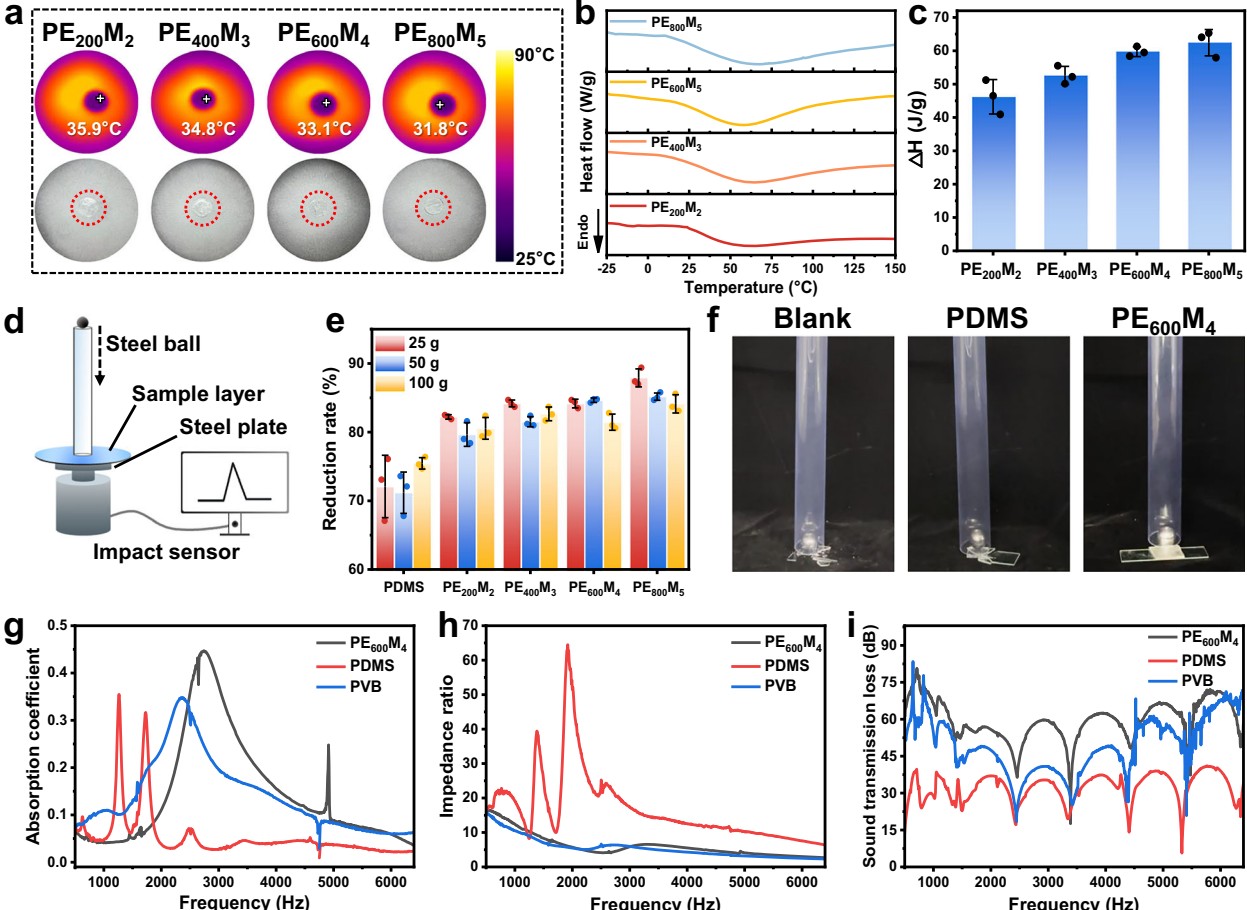

**Fig. 4 | Thermal management, impact protection, and acoustic absorption enabled by the oligomeric-solvent engineering strategy. a** Thermal infrared images and photographs of all $PE_xM_y$ gels after 10 min of exposure to a xenon lamp. **b** DSC curves of all $PE_xM_y$ gels. **c** Summary of enthalpy values of all $PE_xM_y$ gels. **d** Schematic of the falling-ball impact test apparatus. **e** Summary of impact force reduction rates for steel balls of different weights dropped from a fixed height of 50 cm. **f** Photographs showing the impact of a 25 g steel ball dropped from a height of 50 cm onto glass with and without 3-mm-thick protective coating layers. **g**–**i** Sound absorption coefficient, impedance ratio, and sound transmission loss of the $PE_{600}M_4$ gel, PDMS, and PVB samples. Data are presented as mean values ± SD, n = 3 independent samples. Source data are provided as a Source Data file.

gels, attributed to the energy required for dissociation of the hierarchical hydrogen bonds (Fig. 4b). Integration of the DSC curves yielded the corresponding enthalpy values (Fig. 4c), which increased as the MAA content increased, consistent with the higher density of hydrogen bonds formed. To evaluate thermal reversibility, cyclic DSC measurements were conducted using the $PE_{600}M_4$ gel as an example (Supplementary Fig. 28). The heating scan shows a clear endothermic peak, whereas no corresponding exothermic peak appears upon cooling. This behavior reflects the intrinsic nature of the hierarchical hydrogen-bonding network: the endothermic peak originates from the disruption of hierarchical hydrogen-bonding coupled with chain rearrangement. Once they are thermally dissociated, their reformation during cooling is kinetically hindered. Although hydrogen bonds can reform, the slow relaxation dynamics and chain-entanglement constraints prevent recovery of the original organization, resulting in the absence of a distinct exothermic peak.

The excellent damping properties of $PE_xM_y$ gels also make them well suited for impact protection. Their ability to reduce impact force was evaluated using a falling-ball impact test (Fig. 4d), in which steel balls of different weights (25, 50, and 100 g) were dropped freely from a fixed height of 50 cm. Measurement of the impact forces showed that all $PE_xM_y$ gels significantly attenuated the impact of the balls on a steel plate (Supplementary Fig. 29). For example, with the 25 g ball, the

polydimethylsiloxane (PDMS) film reduced the impact force by 72.1%, whereas all $PE_xM_y$ gel films reduced it by more than 82.2% (Fig. 4e), demonstrating the high energy absorption of $PE_xM_y$ gels. We further examined protective performance using the $PE_{600}M_4$ gel. A 25 g steel ball was dropped from 50 cm onto a glass slide covered with either PDMS or $PE_{600}M_4$ gel films. Both the unprotected and PDMS-protected glass fractured, whereas the gel-protected glass remained intact (Fig. 4f and Supplementary Movie 1). To visualize the damping effect, PDMS and $PE_{600}M_4$ balls (diameter = 2 cm) were dropped from 30 cm. The elastic PDMS ball rebounded strongly, while the viscoelastic $PE_{600}M_4$ gel showed no rebound (Supplementary Movie 2). In addition to impact resistance, long-term stability is essential for protective applications. Since $PE_xM_y$ gels contain PEG solvent, rheological tests were conducted to assess possible solvent leakage under high pressure[50]. Samples were held at 50 °C under a normal force of 10 N, with the dynamic modulus recorded every 5 s. After 500 s under continuous pressure, the samples gradually flattened, but no solvent leakage was observed (Supplementary Fig. 30a). The normalized modulus $(G'_x/G'_O)$ slightly increased over time (Supplementary Fig. 30b), confirming the structural stability of the gels under sustained stress.

The damping capability of the gel also extends to the acoustic field. When vibration or noise is transferred from the matrix to the gel,

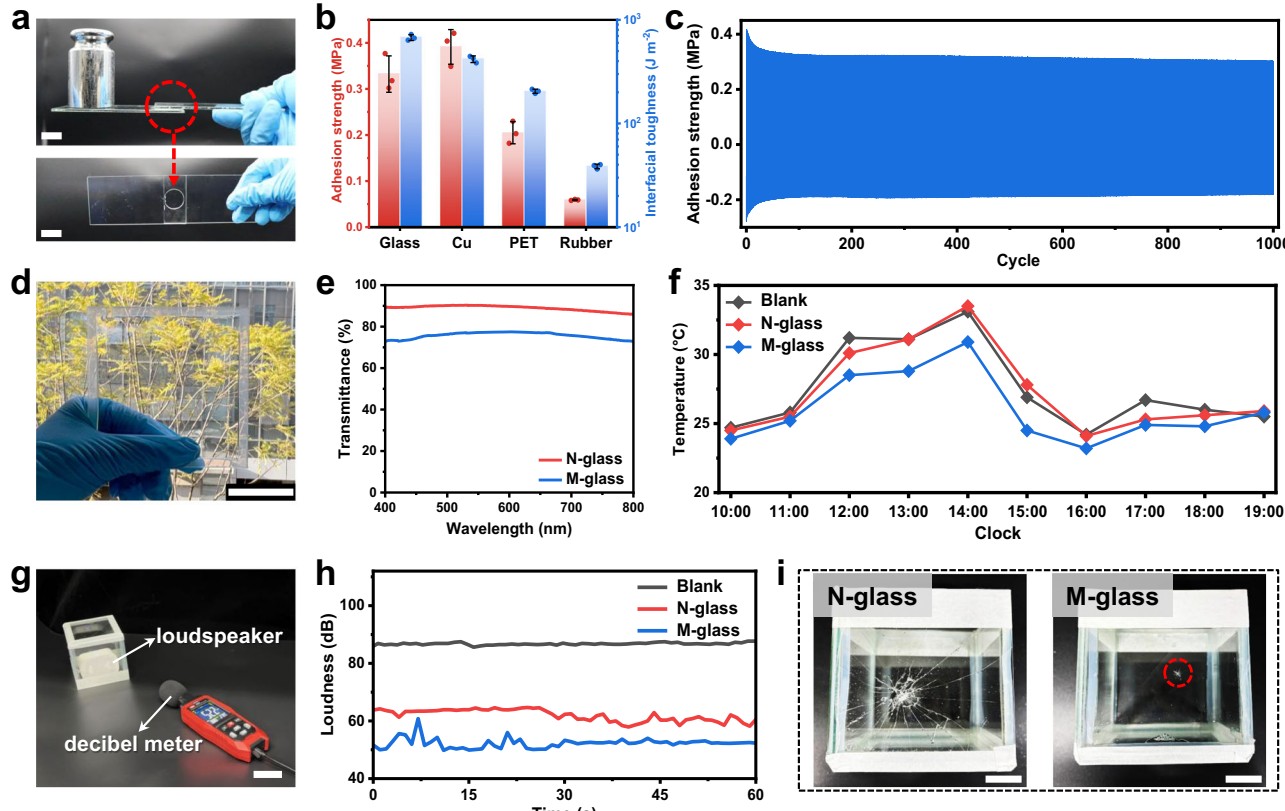

**Fig. 5 | Multifunctional laminated glass composite enabled by oligomeric-solvent-containing PMAA gels. a** Photographs showing the interlayer region (circled) remaining intact during lifting of a 500 g weight. Scale bar: 2 cm. **b** Summary of adhesion strength and interfacial toughness of $PE_{600}M_4$ gel samples on different substrates. **c** Lap-shear stress curve during the repeated loading-unloading cycles. **d** Photograph of the M-glass, showing trees clearly visible behind it. Scale bar: 4 cm. **e** Transmittance spectra of normal glass (N-glass) and M-glass.

**f** Temperature profiles of the blank group, N-glass room, and M-glass room recorded from 10:00 to 19:00. **g** Experimental setup for sound measurement. Scale bar: 2 cm. **h** Loudness variation of the blank group, N-glass room, and M-glass room. **i** Photographs showing the glass rooms after the impact of a 100 g steel ball dropped from a height of 30 cm. Scale bar: 5 cm. Data are presented as mean values ± SD, n = 3 independent samples. Source data are provided as a Source Data file.

mechanical vibration is dissipated through the motion of polymer chains and chain segments, where intermolecular friction converts the mechanical energy into heat, producing a damping effect. To evaluate acoustic noise reduction, the $PE_{600}M_4$ gel was compared with PDMS and polyvinyl butyral (PVB, a commonly used acoustic film). All materials were prepared as disc samples (diameter = 29 mm, thickness = 8 mm) and tested using an acoustic impedance tube. As shown in Fig. 4g, the $PE_{600}M_4$ gel exhibits excellent broadband sound absorption across 2000–6400 Hz, with absorption coefficients higher than those of PDMS and PVB. In particular, it is most responsive in the 2000–4000 Hz range, reaching a peak absorption coefficient of 0.45 near 3000 Hz. This range largely overlaps with typical environmental noise, offering effective protection for hearing health. Additionally, the impedance ratio of the $PE_{600}M_4$ gel is much less than 1, indicating strong reflection and limited transmission of sound (Fig. 4h). Sound attenuation tests further demonstrated that the $PE_{600}M_4$ gel effectively reduces noise intensity across 2000–6400 Hz, with an average reduction of 60 dB, compared to 42 dB for PVB and 30 dB for PDMS (Fig. 4i).

## Application as multifunctional glass interlayers
Windows have long been a key architectural component. In modern buildings, however, they have evolved from simple openings for ventilation and light into multifunctional systems, whose performance directly influences comfort, energy efficiency, and security. High

transparency enhances indoor light transmission and reduces reliance on artificial lighting. Heat absorption lowers energy consumption by reducing thermal load. Sound insulation improves the indoor acoustic environment by blocking external noise. Impact resistance protects against glass breakage in high-rise buildings caused by wind pressure or accidental impact, thereby ensuring safety. The $PE_{600}M_4$ gel integrates all of these properties, making it highly suitable as an interlayer for multifunctional architectural glass, hereafter referred to as M-glass (Supplementary Fig. 31). The preparation of M-glass and N-glass is shown in Supplementary Fig. 32.

As glass interlayers, strong glass–interlayer adhesion is essential for the long-term integrity and multifunctional stability of safety glass. Owing to the abundance of hydroxyl groups, the $PE_xM_y$ gel exhibits robust adhesion. When polymerized in situ on glass, the resulting adhesion is too strong to separate the two substrates. To demonstrate this strong adhesion, a circular gel (diameter = 2 cm, thickness = 1 mm) was polymerized in situ between two glass substrates. Upon application of a 500 g load, no adhesive failure was observed (Fig. 5a and Supplementary Movie 3). The laminated structure was capable of lifting a 2 kg weight. The adhesive joint remained fully intact, while fracture occurred in the glass substrate. To avoid asymmetric adhesion arising from differences in polymerization environments at the two interfaces[51], $PE_{600}M_4$ gels were polymerized between two PET release films. Adhesive performance was quantitatively evaluated using 90° peel and lap-shear tests. The $PE_{600}M_4$ gel demonstrated strong

adhesion on various substrates (Supplementary Fig. 33), achieving an adhesion strength and interfacial toughness on glass as high as 332 kPa and 676 J m$^{-2}$ (Fig. 5b), respectively. In addition, the gel also exhibited strong yet compliant adhesion to human skin (Supplementary Fig. 34). Cyclic shear of the PE$_{600}$M$_4$ gel within ±5% strain generated stable dynamic stress responses. The gel could sustain at least 1000 cycles of lap-shearing deformations (Fig. 5c).

M-glass is highly transparent, with clear visibility of objects behind it (Fig. 5d). UV-vis measurements confirmed that its light transmittance remains above 70% (Fig. 5e), satisfying building glass requirements, though slightly lower than N-glass due to absorption by the gel. To simulate practical use, M-glass and N-glass were assembled into glass chambers for comparative testing. First, their thermal management performance was assessed (Supplementary Fig. 35). A thermometer was placed inside each chamber, while a separate device monitored the blank control group. Temperatures were recorded hourly from 10:00 to 19:00. As shown in Fig. 5f, the temperature inside the M-glass room was consistently lower than in the N-glass and blank groups. When the ambient temperature was below 25 °C, differences between groups were minimal, but above 25 °C, M-glass displayed pronounced heat absorption. At 2:00 pm, the temperature in the M-glass chamber was 2.6 °C lower than outside, consistent with the DSC results (Fig. 4b), which revealed an endothermic peak at around 25 °C. Thus, M-glass effectively lowers indoor temperature to a more comfortable level. Sound reduction was next evaluated (Fig. 5g and Supplementary Movie 4). A loudspeaker played white noise at an average of 87 dB for 60 s in the absence of any glass room. With N-glass installed, the average loudness dropped to 63 dB, while M-glass further reduced it to 51 dB (Fig. 5h), indicating enhanced noise-blocking capability. Finally, impact resistance was tested by dropping a 100 g ball from a height of 30 cm onto the glass rooms. As shown in Fig. 5i and Supplementary Movie 5, N-glass shattered completely upon impact, whereas M-glass sustained only minor localized damage, confirming its protective function. High-speed imaging shows the PE$_{600}$M$_4$ gel protects glass from impact by dissipating stress at the outer layer, preserving the glass's structural integrity (Supplementary Fig. 36 and Supplementary Movie 6).

To demonstrate the practical applicability of M-glass, we subjected it to representative environmental stressors: UV irradiation to simulate solar exposure, cyclic heating to mimic high-temperature conditions, and water rinsing to emulate rainfall. Under all conditions, the M-glass consistently maintained high optical transparency (Supplementary Fig. 37), effective heat absorption, sound-insulation capability (Supplementary Fig. 38 and Supplementary Movie 7), and robust impact resistance (Supplementary Fig. 39 and Supplementary Movie 8), demonstrating excellent stability and functional reliability under realistic environmental scenarios.

Conventional PVB and EVA interlayers often require high-pressure autoclaves or irreversible thermal crosslinking to achieve optical transparency; without such treatment, trapped air, crystallinity, or phase inhomogeneity causes light scattering. Instead, we emphasize that the proposed gel-based interlayer represents a distinct and complementary strategy for laminated glass fabrication, prioritizing manufacturing simplicity, optical reliability, interfacial adaptability, and processing flexibility (Supplementary Table 2).

## Discussion

In this paper, we developed a simple gel system featuring a hierarchical hydrogen-bonding network. PEG oligomers serve as molecular bridges between PMAA chains, promoting the formation of a multilevel hydrogen-bonding framework. This design imparts the gel with the combined advantages of elastic solids and viscous fluids: it can dissipate strain energy effectively while maintaining structural integrity and avoiding free flow. Consequently, the gel exhibits good damping performance, with damping capacity above 80% and a loss factor

approaching 1 across a broad frequency range. By converting external mechanical energy into internal dissipation, the gel is well suited for impact protection and sound absorption. Additionally, the reversible dissociation of hierarchical hydrogen bonds at elevated temperatures produces an endothermic effect, enabling temperature regulation. Harnessing these multifunctional properties, we fabricated a proof-of-concept glass that integrates high transparency, heat absorption, noise reduction, and impact resistance. The successful demonstration of this system highlights the strong potential of gel-based glass interlayers in advancing sustainable, low-carbon, and safe living environments.

## Methods

### Materials

Methacrylic acid (MAA), polyethylene glycol (PEG, molecular weight of 200, 400, 600, and 800), acrylic acid (AAc), and 2-hydroxy-2-methylpropiophenone (HMPP) were provided by Aladdin Biochemical Technology Co., Ltd. (Shanghai, China). N,N'-methylene bisacrylamide (MBAA) was purchased from Kelong Chemical Reagent Factory (Chengdu, China). PVB tape was purchased from Xiangrui Factory (Jiangsu, China). Glue A and Glue B (Dow Corning DC184) used for the preparation of PDMS elastomers were supplied by Dow Corning Co. Ltd (Shanghai, China). All reagents were used as received without further purification.

### Preparation of PE$_x$M$_y$ gels

The gels were prepared through one-pot free-radical polymerization. As a representative procedure, PEG-200 (0.01 mol), MAA (0.018, 0.02, 0.022, or 0.024 mol), MBAA (0.1 mol% relative to MAA), and HMPP (0.2 mol% relative to MAA) were mixed as the precursor solution. The precursor was injected into a PTFE reaction cell and polymerized under UV irradiation (365 nm, 60 W) for 45 min. The resulting gels were denoted as PE$_x$M$_y$, where $x$ represents the molecular weight of PEG, and $y$ represents the molar ratio of MAA to PEG. Unless otherwise specified, the PE$_{200}$M$_y$ gels (abbreviated as PEM$_y$) were used as the representative samples for tests and characterizations. For comparison, pure PMAA was prepared by bulk polymerization, a PMAA hydrogel was obtained by replacing PEG with H$_2$O, and a PEA gel was synthesized by substituting MAA with AAc, while keeping all other polymerization conditions identical. The detailed compositions are summarized in Supplementary Table 1.

### Preparation of PDMS elastomer

PDMS elastomer was synthesized by mixing Glue A and Glue B at a mass ratio of 10:1, followed by thorough degassing to remove air bubbles. The mixture was cast into a PTFE mold and cured in an oven at 60 °C for 3 h to obtain the PDMS elastomer.

### Characterizations

Transmittance spectra of approximately 2 mm gel films were recorded on a Shimadzu UV-Vis spectrophotometer. Fourier transform infrared (FTIR) spectroscopy spectra were acquired on a Thermo Fisher Scientific spectrometer (USA); gel samples were measured in attenuated total reflectance (ATR) mode, and liquid samples were prepared using KBr pellets. Variable-temperature IR spectra (30-150 °C, 3 °C min$^{-1}$) were collected in transmission mode (Nicolet iS10, Thermo Fisher Scientific, USA) using ZnSe windows. $^1$H nuclear magnetic resonance (NMR) spectra were recorded on a Bruker spectrometer (Germany) at 298 K using D$_2$O as solvent. Gel microstructures were observed using a Thermo Fisher Scientific scanning electron microscope (SEM). Thermal gravimetric analysis (TGA) was performed on a NETZSCH analyzer (Germany) from 45-800 °C at 10 °C min$^{-1}$ under N$_2$ atmosphere. Differential scanning calorimetry (DSC) curves were obtained using a Mettler calorimeter (USA) from -30-150 °C at 5 °C min$^{-1}$ under N$_2$ atmosphere. Dynamic mechanical analysis (DMA) was performed in tensile mode at 0.1% strain from -20-150 °C at 3 °C min$^{-1}$. Small-angle X-

ray scattering (SAXS) measurements were performed on a Xeuss system (France) with 50 keV X-ray energy (wavelength: 1.54189 Å), and a sample-to-detector distance of 2841.36 mm. 2D SAXS patterns were recorded on a PILATUS3 300 K detector and analyzed using Foxtrot Academic Edition software. Rheology tests were conducted using an Anton Paar rheometer (Austria). Frequency sweeps (1-100 Hz) were performed at 1% strain, with a normal force of 5 N. Temperature was varied from 5-95 °C at 10 °C min$^{-1}$. Master curves at 25 °C were generated via time-temperature superposition.

## Molecular dynamics (MD) simulation

An amorphous cell containing 5 PMAA chains and 50 PEG molecules was constructed. Simulations were performed with the COMPASS II force field in the Forcite module (Materials Studio). The cell was annealed five times (300-500 K) under NVE conditions, followed by 5 ns of NPT simulation (298 K, 1 atm, with a 5 fs timestep). The maximum hydrogen-acceptor distance was set to 2.5 Å for hydrogen bond analysis.

## Tensile testing

Gel samples polymerized in dumbbell-shaped molds ($16 \times 4 \times 3$ mm$^3$) were tested using an Instron universal testing machine (USA) with a 500 N load cell. Uniaxial tensile tests were conducted at 100 mm min$^{-1}$ unless otherwise specified. Young's modulus was calculated from the initial linear region of stress-strain curves. Rate dependence was examined at various tensile velocities, and temperature dependence was examined at different test temperatures. For cyclic loading-unloading, dissipated energy (hysteresis) was determined by integrating the area between loading and unloading curves.

## Single-edge notch test

Rectangular gel samples ($20 \times 5 \times 3$ mm$^3$) were prepared. Notched samples contained a 1 mm central notch, while unnotched samples were intact. Uniaxial tensile tests were run at room temperature at 100 mm min$^{-1}$. Fracture energy was calculated using[52]:

$$G = \frac{6}{\sqrt{\lambda}} W(\lambda_b) c \qquad (3)$$

where $c$ is the notch length, $\lambda$ is the strain at crack initiation in the notched sample, and $W(\lambda_b)$ is the strain energy density of the unnotched sample at the same strain.

## Stress relaxation test

Dumbbell-shaped samples ($16 \times 4 \times 3$ mm$^3$) were stretched to a fixed strain (1 mm/mm) and held for 10 min. Stress relaxation followed:

$$\sigma = \sigma_0 e^{-\frac{t}{\tau}} \qquad (4)$$

where $\sigma_0$ is the initial stress and $\tau$ is the relaxation time.

## Large amplitude oscillatory shear (LAOS) tests

The LAOS tests were carried out using an Anton Paar rheometer at 25 °C. Under an applied sinusoidal stress input ($\sigma = \sigma_0 \sin \omega t$), the gels exhibited a corresponding strain response ($\varepsilon = \varepsilon_0 \sin(\omega t - \delta)$), where $\delta$ denotes the phase lag. The enclosed area of the resulting stress-strain hysteresis loop reflects the energy dissipation per cycle.

## Falling-ball test

Impact resistance was measured using a falling-ball tester (LICHEN, China). Steel balls (25, 50, and 100 g) were dropped from 50 cm onto gel-covered target discs, and the resulting impact force was recorded.

## Acoustic testing

Acoustic performance was evaluated with an impedance tube (4206-T, HBK, UK) in the frequency range of 500-6400 Hz. Disc samples (29 mm in diameter and 8 mm thick) were tested under normal incidence. The sound absorption coefficient, impedance ratio, and transmission loss were measured.

## Adhesion testing

Adhesion properties were evaluated by 90° peel and lap-shear tests using an Instron universal testing machine at room temperature. For the 90° peel test, gel samples ($25 \times 15 \times 1$ mm$^3$) backed with polyimide film were adhered to the substrates at a peel rate of 100 mm min$^{-1}$. For lap-shear tests, gel samples ($10 \times 5 \times 1$ mm$^3$) were sandwiched between two substrates to form a laminated structure and loaded at a testing speed of 100 mm min$^{-1}$. To demonstrate the adhesion performance of the gel, the $PE_{600}M_4$ gel was temporarily attached to the experimenter's hand and then carefully removed. The surface morphology of the gel was subsequently examined under an optical microscope (Leica, Germany). The hand shown in Supplementary Fig. 34 belongs to one of the authors. Written informed consent for both participation in the experiment and publication of the image was obtained.

## Cyclic lap-shear test

Cyclic lap-shear tests were performed on gel samples ($20 \times 50 \times 1$ mm$^3$) sandwiched between two glass substrates. The tests were conducted at a loading rate of 100 mm min$^{-1}$ with the shear strain controlled within ±5%.

## Data availability

The data supporting the findings of this study are available within the Article and its Supplementary Information. All raw data are available from the corresponding author on request. Source data are provided with this paper.

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

## Acknowledgements

This work was supported mainly by the National Natural Science Foundation of China (52473238 (W.C.), 52203027 (W.C.), 52573231 (R.R.)), the Sichuan Science and Technology Program (2024NSFSC0245 (W.C.), 2025ZNSFSC0340 (R.R.)), the State Key Laboratory of Polymer Materials Engineering (sklpme-2024-01-04 (R.R.)), and the State Key Laboratory of Advanced Polymer Materials (sklamp2025-2-04 (W.C.)). The authors also gratefully acknowledge the support from the Sichuan Province Advanced Building Materials Production-Education Integration Innovation Demonstration Platform.

## Author contributions

R.R. and W.C. conceived and supervised the project and designed the experiments. M.L. performed the majority of the experiments. L.H., X.Y.,

and X.H. assisting in sample synthesis and M.P. supporting the Molecular Dynamics simulations. M.L., R.R., and W.C. co-wrote and revised the paper. All authors discussed the results and commented on the manuscript.

## Competing interests

The authors declare no competing interests.
