## [Transparent Peer Review file · Nature Communications]

Oligomeric-Solvent Engineering of Hierarchical Hydrogen-bonding Networks for Multifunctional Glass Interlayers

Corresponding Author: Professor Rong Ran

Version 0:

Reviewer comments:

Reviewer #1

(Remarks to the Author)

In this work, the authors introduce oligomeric polyethylene glycol (PEG-200) as a “solvent” for poly(methacrylic acid) (PMAA) to engineer hierarchical hydrogen-bond networks that couple elasticity and viscosity. They argue this architecture yields transparent, dissipative gels with (i) room-temperature broadband viscoelastic damping ($\tan \delta \approx 1$ over many decades of frequency via TTS), (ii) impact resistance, (iii) thermal buffering due to endothermic H-bond dissociation, (iv) adhesion/self-healing, and (v) optical clarity. These combined properties enable application as a multifunctional laminated glass interlayer, and the concept is further extended to higher-molecular-weight PEGs by adjusting composition to maintain gel-like behavior. The study is promising and near the bar for publication in this journal. However, several aspects require clarification, additional quantification, and expanded controls to strengthen the mechanistic claims and application relevance:

1. While the concept of oligomeric solvents is novel, several recent works (e.g., on ion-dipole hybrid gels or deep eutectic solvent gels) also report hierarchical interactions. The authors should explicitly differentiate their approach in terms of molecular design strategy rather than performance alone.
2. PEG-200 is framed as a “solvent.” Given it is a low-MW polymeric liquid with strong H-bonding to PMAA, many readers will view it as a plasticizer/cosolvent. The authors should clearly define how this “oligomeric solvent” differs from traditional small-molecule solvents such as water, particularly in terms of molecular interaction, volatility, and hydrogen-bonding behavior. Comparative experiments with water-based PMAA gels would clarify this distinction.
3. The authors mention that the concept extends to higher-MW PEGs, yet the influence of PEG molecular weight on the hierarchical hydrogen-bond architecture is unclear. Please clarify how varying PEG chain length affects hydrogen-bond density, phase homogeneity, and mechanical behavior. Comparative FTIR or rheological data could strengthen this section.
4. The hierarchical hydrogen-bond network is central to the paper, yet the supporting data (FTIR, DSC, MD simulations) are largely indirect. Incorporating more quantitative validation such as temperature-dependent NMR, 2D-IR or other spectroscopic techniques would strengthen the mechanistic claim of multi-tiered hydrogen bonds.
5. It remains uncertain whether the observed endothermic transitions are reversible. Please include cyclic DSC results to confirm the reversibility of hydrogen-bond dissociation and reassociation during heating-cooling cycles. This would provide stronger evidence for the proposed thermal buffering mechanism.
6. The demonstration of the “M-glass” system is interesting, yet its long-term stability under realistic conditions (humidity, UV exposure, and thermal cycling) has not been discussed. Such data—or at least an informed discussion—are essential to evaluate its suitability for real-world glazing applications.
7. For the falling-ball tests and glass-room drop tests, please add: replicates, error bars, and statistical treatment of impact force reductions.
8. Common glass interlayers include PVB and EVA. Please provide matched-thickness benchmarking against these commercial materials to contextualize performance. Clarify failure modes of the composite glass (glass cracking patterns, gel deformation) using high-speed images or post-impact observations.

Reviewer #2

(Remarks to the Author)

This manuscript presents a PEG oligomer-based gel with functionalities in thermal regulation, sound attenuation, and mechanical protection. While the reported properties are interesting, the core concept — polymer oligomer solvent-based gels — is not novel. PEG-based “PEGgels” and related systems have been extensively studied in recent years, with detailed

mechanistic, mechanical, and functional analyses already published (e.g., Nat. Commun. 2021, 12, 3610; Adv. Mater. 2022, 34, 2107791; Sci. Adv. 2025, 11, eadv5292). These works already have covered thermal stability, tunable mechanics, self-healing, energy dissipation, and protective applications. The authors highlight thermal regulation and glass interlayer application as new angles, but the overall novelty of this work requires further clarification based on the previous work. In other words, the use of PEG oligomers as solvent/structural components in gels is well-established. The manuscript does not convincingly demonstrate how this system advances beyond existing PEGgel literature, particularly in terms of mechanism or application scope. Therefore, I don't think the manuscript in current form is ready for publication in Nat. Commun..

Some other issues are as below:

1. The comparison to starfish stiffness modulation is misleading. Starfish employ complex, reversible, biologically regulated systems (blood pressure, peptides); the PEM gel simply softens via PEG plasticization — a fundamentally different and far less sophisticated mechanism.
2. Page 12, Line 261. The explanations about elasticity and viscosity (consistent damping factor) across broad temperatures lack scientific theories or experiment support, authors can refer to some previous work (e.g. Nat. Commun. 2021, 12(1), 3610).
3. Figure 3h. Typically, to evaluate the damping capacity of materials at varied tensile strains, the tensile tests are performed at separated strain cycles instead of consecutive cycles.
4. The explanations on the spectroscopy should be more comprehensive. How is the formation of H-bonds between PEG and MAA resulting in the blueshift? What about the corresponding shift for PEG.
5. The manuscript introduces asymmetric adhesion without explaining its functional purpose, to me it appears to offer only drawbacks.
6. Missing figure captions and wrongly assigned figure numbers for figures S8 and after.

Reviewer #3

(Remarks to the Author)

The manuscript by Li et al. reports a polymeric material composed of poly(methacrylic acid) (PMAA) swollen in poly(ethylene glycol) (PEG), forming a gel-like network that exhibits properties including high energy dissipation, acoustic damping, self-healing, adhesiveness, and impact resistance. These properties arise from the interaction between PEG macromolecules and the PMAA network. The material and its performance are well characterized, and the authors demonstrate several proof-of-concept applications. Overall, the experimental work is solid and the results are of potential interest.

However, the presentation of prior work is incomplete and somewhat misleading. The concept of using PEG as a macromolecular solvent was first introduced by Wang Z. et al. (Adv. Mater. 2022, "Tough, Transparent, 3D-Printable, and Self-Healing Poly(ethylene glycol)-Gel (PEGgel)"), where the fundamental principles, transparency, self-healing, and exceptional mechanical properties were already demonstrated. Subsequent studies, including Wang M. et al., Nature 2024, have further explored applications and properties of PEG-polymer hybrid materials.

The authors should therefore revise the Abstract and Introduction to properly acknowledge this prior work and avoid implying that the present study introduces the concept for the first time (e.g., phrases such as "our oligomeric-solvent engineering strategy" or "we introduce oligomeric PEG into a PMAA network to construct a gel system that addresses key limitations of traditional solvent-based gels" are misleading).

With appropriate correction of the contextual framing, this manuscript would represent a valuable contribution to the field highlighting the advantages and further applications of an already established concept.

Version 1:

Reviewer comments:

Reviewer #1

(Remarks to the Author)

The revised manuscript is suitable for publication.

Reviewer #2

(Remarks to the Author)

The authors have thoroughly addressed the reviewers' critiques by significantly enhancing the mechanistic evidence through additional FT-IR, 2D-COS, and MD simulation data, which explicitly validate the hierarchical hydrogen-bonding network. They have also clearly acknowledged relevant prior work and differentiated their work from prior PEG-gel studies by emphasizing the single-interaction-family design and broadened damping performance, supported by new environmental stability tests and quantitative impact analyses. The revisions have strengthened the manuscript's novelty, data completeness, and practical relevance, making it suitable for publication.

Dear Reviewers,

Thank you very much for reviewing our manuscript entitled “Bioinspired Oligomeric-Solvent Engineering of Hierarchical Hydrogen-Bond Networks for Multifunctional Glass Interlayers” (Manuscript ID: NCOMMS-25-73988). We sincerely appreciate your time and the constructive comments provided.

In response to your valuable suggestions, we have carefully revised the manuscript and made substantial improvements. All revisions have been highlighted in red in the revised manuscript, and detailed point-by-point responses to each comment are provided below.

Thank you again for your insightful feedback, which has greatly helped us improve the quality and clarity of the manuscript. We believe that the revised version is significantly improved compared to the original submission.

Sincerely,

Rong Ran

Response to the reviewers' comments

Reviewer #1:

In this work, the authors introduce oligomeric polyethylene glycol (PEG-200) as a “solvent” for poly(methacrylic acid) (PMAA) to engineer hierarchical hydrogen-bond networks that couple elasticity and viscosity. They argue this architecture yields transparent, dissipative gels with (i) room-temperature broadband viscoelastic damping ($\tan \delta \approx 1$ over many decades of frequency via TTS), (ii) impact resistance, (iii) thermal buffering due to endothermic H-bond dissociation, (iv) adhesion/self-healing, and (v) optical clarity. These combined properties enable application as a multifunctional laminated glass interlayer, and the concept is further extended to higher-molecular-weight PEGs by adjusting composition to maintain gel-like behavior. The study is promising and near the bar for publication in this journal. However, several aspects require clarification, additional quantification, and expanded controls to strengthen the mechanistic claims and application relevance:

Response: We appreciate the reviewer for this positive and insightful assessment of our manuscript. In response to the reviewer's comments, we have thoroughly revised the Manuscript and Supporting Information to strengthen the mechanistic interpretation of the hierarchical hydrogen-bonding network,

provide additional quantitative analyses where appropriate, and expand control experiments to better substantiate the structure–property–function relationships.

Below, we address each specific point raised by the reviewer in detail.

1. While the concept of oligomeric solvents is novel, several recent works (e.g., on ion-dipole hybrid gels or deep eutectic solvent gels) also report hierarchical interactions. The authors should explicitly differentiate their approach in terms of molecular design strategy rather than performance alone.

Response: We thank the reviewer for highlighting this important point. We agree that hierarchical interactions have been reported in several recent gel systems, including ion–dipole hybrid gels and deep eutectic solvent–based gels. However, we emphasize that the central novelty of our work lies in the molecular design strategy, rather than in the observation of hierarchical interactions per se.

Specifically, the hierarchical hydrogen-bonding network in our system does not arise from mixing multiple types of strong interactions (e.g., ionic coordination, ion–dipole coupling, or eutectic complexes). Instead, it emerges from a single interaction motif—hydrogen bonding—whose strength distribution is broadened by the introduction of an oligomeric solvent with finite chain length. In this oligomeric-solvent strategy, PEG does not function as a functional additive or co-network former, but as an oligomeric solvent that simultaneously (i) dilutes and spatially separates PMAA chains, (ii) bridges neighboring chains through multidentate hydrogen bonding, and (iii) preserves chain entanglement and mobility over extended length scales.

This design principle is fundamentally distinct from ion-dipole or eutectic gel approaches, where hierarchy is typically introduced through chemical heterogeneity (multiple bond chemistries, ions, or eutectic components). In contrast, our approach achieves hierarchy through topological and length-scale modulation of a single hydrogen-bonding chemistry, enabled by the oligomeric nature of the solvent. This difference leads to a qualitatively different viscoelastic landscape, characterized by a continuous spectrum of relaxation times rather than discrete relaxation modes.

To clarify this distinction, we have revised the Introduction and Discussion to explicitly compare these design strategies and to articulate how oligomeric-solvent engineering represents a complementary, conceptually simpler route to hierarchical networks. The updated discussion revised on page 8 of the manuscript.

Page 8: *“It is worth noting that hierarchical interactions have also been reported in several recent gel*

systems³⁸⁻⁴⁰, such as ion–dipole hybrid gels and deep eutectic solvent–based gels. In these materials, hierarchy is typically introduced through chemical heterogeneity, for example by combining multiple interaction motifs (ionic coordination, ion–dipole coupling, or eutectic complexation) with distinct bonding energies. By contrast, we adopt a fundamentally different molecular design strategy in which hierarchical behavior emerges from a single interaction type—hydrogen bonding—whose strength distribution is continuously broadened through oligomeric-solvent engineering. The finite chain length of oligomeric PEG enables it to function simultaneously as a solvent, spacer, and multidentate hydrogen-bonding bridge, generating hierarchy through topological and length-scale modulation. This strategy provides a conceptually simple and general route to hierarchical viscoelastic networks.”

2. PEG-200 is framed as a “solvent.” Given it is a low-MW polymeric liquid with strong H-bonding to PMAA, many readers will view it as a plasticizer/cosolvent. The authors should clearly define how this “oligomeric solvent” differs from traditional small-molecule solvents such as water, particularly in terms of molecular interaction, volatility, and hydrogen-bonding behavior. Comparative experiments with water-based PMAA gels would clarify this distinction.

Response: We thank the reviewer for raising this important point. PEG functions as an oligomeric solvent, which is fundamentally different from conventional small-molecule solvents such as water. In this work, PEG oligomeric solvent serves two functions. First, owing to their oligomeric nature, PEG chains possess a substantially greater contour length than small-molecule solvents and can more effectively separate adjacent PMAA chains, thereby increasing interchain spacing and free volume. Second, the ether oxygen atoms along the PEG backbone bear lone pairs and act as hydrogen-bond acceptors, forming multiple bidentate (medium-strength) and monodentate (weak) interactions with the carboxylic groups of PMAA. As a result, the introduction of oligomeric PEG broadens the hydrogen-bond strength distribution and enables the construction of a hierarchical hydrogen-bonding network.

To illustrate this distinction, we performed a comparison in which PEG-200 was replaced by water at the same molar ratio (e.g., PEM₂: 0.02 mol MAA and 0.01 mol PEG or water). Because of the much smaller molecular weight of water, this substitution corresponds to the addition of only 0.18 g of water. Under these conditions, the resulting material shows little difference from bulk polymerized PMAA and remains a white, hard, and brittle plastic sheet (**Fig. S1**). This result indicates that simply matching

the molar ratio cannot reproduce the role played by PEG in the PE_xM_y system. More importantly, even when the water content is increased, water cannot replicate the function of PEG. Although water molecules can form hydrogen bonds with PMAA via their hydroxyl groups, they exist as discrete, non-bridging species. Consequently, water primarily competes for and occupies the carbonyl groups of PMAA, weakening interchain PMAA–PMAA hydrogen bonding rather than constructing a stable bridging network. In contrast, PEG chains can simultaneously interact with PMAA chains, thereby enabling effective chain separation and network connectivity.

Taken together, these results demonstrate that PEG oligomers act as an “oligomeric solvent” that uniquely combines molecular bridging and plasticization effects. This dual functionality fundamentally distinguishes PEG oligomers from traditional small-molecule solvents such as water. The updated results are now presented in **Fig. S1**, with the corresponding discussion revised on **page 5** of the manuscript.

Fig. S1. Photographs show that substituting PEG with H_2O at the same molar ratio (PMMA hydrogel: 0.02 mol MAA and 0.01 mol H_2O) produces a white, hard, and brittle plastic sheet rather than a hydrogel. This result indicates that matching the molar ratio alone is insufficient to reproduce the role of PEG in the PE_xM_y system. H_2O disrupts PMAA–PMAA hydrogen bonding but fails to establish a stable bridging network, whereas PEG oligomers can simultaneously interact with multiple PMAA chains, enabling effective chain separation and network connectivity.

Page 5: “*Notably, both PEG and PMAA are indispensable for building this hierarchical hydrogen-bonding network. Substituting PEG with H_2O results in a glassy, brittle PMAA material rather than a gel (Fig. S1), while replacing PMAA with polyacrylic acid (PAAc) suppresses gelation in the PEG/PAAc (PEA) system (Fig. S2).*”

3. The authors mention that the concept extends to higher-MW PEGs, yet the influence of PEG molecular weight on the hierarchical hydrogen-bond architecture is unclear. Please clarify how varying PEG chain length affects hydrogen-bond density, phase homogeneity, and mechanical behavior.

Comparative FTIR or rheological data could strengthen this section.

Response: We thank the reviewer for this insightful comment. In our system, increasing the PEG chain length does not compromise the phase homogeneity of the PE_xM_y gels, as evidenced by their consistently high optical transparency (**Fig. S26a**). However, longer PEG chains increase the spatial separation between adjacent PMAA chains, which reduces the proportion of strong, short-range PMAA–PMAA hydrogen bonds while increasing the fraction of relatively weak, long-range PMAA–PEG hydrogen bonds.

To verify this effect, we synthesized PE_xM_2 gels at a fixed MAA/PEG molar ratio ($x = 2$) using PEGs of different molecular weights (400, 600, and 800 g/mol). All formulations ($PE_{400}M_2$, $PE_{600}M_2$, and $PE_{800}M_2$) formed sol-like rather than gel-like materials (**Fig. S25**), indicating longer PEG chains hinder the formation of strong hydrogen bonds. Notably, $PE_{400}M_2$ retains partial mechanical integrity and does not fully flow upon inversion, whereas $PE_{800}M_2$ behaves entirely as a viscous solution. This progressive loss of structural integrity confirms that increasing PEG chain length weakens the strong hydrogen-bond density, shifting the network toward weaker PMAA–PEG interactions.

To obtain self-supporting gels with higher molecular weight PEGs, the MAA content was increased to compensate for the reduced density of strong hydrogen bonds. When adjusted accordingly, $PE_{400}M_3$, $PE_{600}M_4$, and $PE_{800}M_5$ all exhibited robust gel-like behavior. As shown in **Fig. S26b**, their stress–strain curves demonstrate enhanced mechanical strength, confirming that the increased MAA content effectively restores strong hydrogen bonding. Moreover, the damping capacity (**Fig. S26c**) and rheological measurement results (**Fig. S27**) further confirm that the chain length of PEG does not compromise the damping performance of the gel. These results collectively show that, despite variations in PEG chain length, the gels consistently form a hierarchical hydrogen-bonding network, with PEG oligomers serving as molecular bridges. The updated results are now presented in **Fig. S25**, with the corresponding discussion revised on page 13 of the manuscript.

Page 13: *“Furthermore, we also emphasize that our oligomeric-solvent engineering strategy is universal. To examine its generality, PEGs with different molecular weights (400, 600, and 800 g/mol) were employed to prepare PE_xM_y gels. Longer PEG chains act as plasticizers, increasing the distance between PMAA chains and thereby reducing the number of hydrogen bonds formed. As a result, replacing PEG-200 with PEG-400, PEG-600, and PEG-800 in the PEM_2 formulation yielded sol-like rather than gel-like products (**Fig. S25**). To obtain comparable gel states, the MAA content was*

accordingly increased, resulting in the $PE_{400}M_3$, $PE_{600}M_4$, and $PE_{800}M_5$ formulations. All of these gels display high transparency (**Fig. S26a**). Their strain-stress curves confirm robust, gel-like toughness (**Fig. S26b**), and they exhibit excellent damping capacity (**Fig. S26c**). The rheological behavior of $PE_{400}M_3$, $PE_{600}M_4$, and $PE_{800}M_5$ is summarized in **Fig. S27**, showing that all maintain a balance between viscosity and elasticity across an ultra-wide frequency range, with $\tan\delta$ values approaching 1.”

Fig. S25. Photographs of $PE_{400}M_3$, $PE_{600}M_4$, and $PE_{800}M_5$ gels. The $PE_{400}M_2$ sample retains partial mechanical integrity and does not fully flow upon inversion, whereas the $PE_{800}M_2$ sample behaves entirely as a viscous solution. This progressive loss of structural integrity confirms that increasing the PEG chain length weakens the strong hydrogen-bonding density and toward to weaker PMAA–PEG hydrogen-bonding.

4. The hierarchical hydrogen-bond network is central to the paper, yet the supporting data (FTIR, DSC, MD simulations) are largely indirect. Incorporating more quantitative validation such as temperature-dependent NMR, 2D-IR or other spectroscopic techniques would strengthen the mechanistic claim of multi-tiered hydrogen bonds.

Response: We sincerely thank the reviewer for this insightful and constructive comment. We agree that rigorous validation of the proposed hierarchical hydrogen-bonding network is essential. In response, we have incorporated systematic FT-IR analysis combined with temperature-dependent two-dimensional infrared (2D-IR) spectroscopy to directly elucidate the hierarchical hydrogen-bonding network in the gel.

First, the coexistence of hydrogen bonds with distinct strengths was confirmed by FT-IR spectroscopy. Neat MAA exhibits a dominant carbonyl peak at $\sim 1700\text{ cm}^{-1}$ (**Fig. S3a**), which is characteristic of strong dimeric PMAA–PMAA hydrogen bonds. Upon incorporation of PEG, an additional blue-shifted carbonyl peak emerges at $\sim 1728\text{ cm}^{-1}$ (**Fig. S3b–e**). This shift reflects the partial disruption of strong PMAA–PMAA dimers and the formation of weaker PEG–PMAA hydrogen bonds.

Importantly, quantitative peak-area ratio analysis demonstrates that the relative fraction of strongly hydrogen-bonded carbonyl groups increases monotonically with increasing MAA content (**Fig. S3f**), providing direct evidence for the composition-dependent hierarchy of hydrogen-bond interactions within the gel.

Consistent signatures are also observed in the PEG segment. The C–O–C stretching band of PEG (1000–1100 cm^{-1}) exhibits pronounced broadening and increasing asymmetry with rising MAA content, accompanied by the emergence of a low-wavenumber shoulder (**Fig. S4**). This spectral evolution signifies the coexistence of free ether oxygens and hydrogen-bonded ether oxygens, confirming the formation of PEG–PMAA hydrogen bonds and further supporting the heterogeneous hydrogen-bonding environment.

Second, the thermal evolution of the hierarchical hydrogen-bond network was directly probed by temperature-dependent FT-IR and 2D correlation infrared spectroscopy. FT-IR spectra of the PEM₂ gel were collected during heating from 30 to 150 °C. In the O–H stretching region of PMAA (**Fig. 2d**), a gradual blue shift from ~ 3500 to ~ 3515 cm^{-1} is observed, indicating progressive dissociation of hydrogen-bonding aggregates into weaker associated or free carboxyl groups.

To resolve the sequential nature of these thermal events, two-dimensional correlation spectroscopy (2DCOS) was employed. The synchronous spectrum (**Fig. 2e**) reveals cooperative spectral changes across the broad 2700–3600 cm^{-1} region, confirming the collective response of hydrogen-bonded species to thermal perturbation. More critically, the asynchronous spectrum (**Fig. 2f**) distinguishes the order of molecular events: the response of strongly hydrogen-bonded O–H groups (~ 3297 cm^{-1}) precede that of weakly associated or free O–H groups (~ 3596 cm^{-1}), directly evidencing stepwise thermal dissociation of hydrogen bonds with different strengths.

Additional mechanistic insight is obtained from the carbonyl stretching region (1690–1750 cm^{-1} , **Fig. S5**). The synchronous 2D-IR spectrum shows positive cross-peaks between the ~ 1700 cm^{-1} band (dimeric PMAA–PMAA hydrogen bonds) and the ~ 1728 – 1730 cm^{-1} band (free or PEG–PMAA-associated carbonyls), indicating correlated intensity changes upon heating. According to Noda's rule, the presence of both positive synchronous and positive asynchronous cross-peaks at (1700, 1728 cm^{-1}) establishes a well-defined sequence: the disruption of strong PMAA–PMAA hydrogen bonds occurs first, followed by the emergence of weaker or free carbonyl species. This spectroscopic evidence unambiguously confirms the hierarchical and discontinuous nature of the hydrogen-bond network.

Collectively, these results provide direct experimental validation of the proposed multi-tiered hydrogen-bonding mechanism, thereby substantially strengthening the mechanistic foundation of the manuscript. The updated results are now presented in **Fig. 2d-f**, **Fig. S3-S5**, with the corresponding discussion revised on **page 7** of the manuscript.

Fig. 2. Characterizations and simulations revealing hierarchical hydrogen-bonding in PEM_y gels. **d.** Temperature-dependent FTIR spectra of PEM_2 gel upon heating from 30 to 150 °C (10 °C intervals). **e, f.** Synchronous and asynchronous 2DCOS spectra of the PEM_2 gel. Red regions indicate positive intensities, while blue regions represent negative ones.

Fig. S3. Peak fitting analysis of the carbonyl stretching vibration [$\nu(\text{C}=\text{O})$] at 1700 cm^{-1} . **a.** Pure MAA. **b-e.** PEM_y gel, $y = 1.8, 2, 2.2,$ and 2.4 . **f.** Corresponding area fractions of the fitted carbonyl peak.

Figure S4. Enlarged FT-IR spectra in the C–O–C stretching region (1000–1100 cm^{-1}) of neat PEG and PEM_y gels.

Fig. S5. Synchronous and asynchronous 2DCOS spectra of the PEM_2 gel. Red regions indicate positive intensities, while blue regions represent negative ones.

Page 7: “Fourier transform infrared (FT-IR) spectroscopy further elucidate the chemical composition and interaction mechanism within the PEM_y gels (Fig. 2c). For neat PEG, absorption peaks at 2876 cm^{-1} and 2930 cm^{-1} correspond to symmetrical and asymmetric stretching of $-\text{CH}_2$ groups, while the peak at 1059 cm^{-1} is assigned to C–O–C stretching^{33,34}. For neat MAA, the characteristic C=O stretching vibration appears near 1700 cm^{-1} . These characteristic peaks are present in all PEM_y gel samples, confirming the successful incorporation of both components.

To clarify the hierarchical nature of the hydrogen-bonding network, peak deconvolution was performed on the C=O vibration FT-IR spectra. Neat MAA exhibits a prominent carbonyl peak at 1700 cm^{-1} (Fig. S3a), characteristic of strong dimeric hydrogen bonds. Upon PEG incorporation, an additional blue-shifted peak emerges near 1728 cm^{-1} (Fig. S3b-e). This shift arises from the disruption of partial strong PMAA–PMAA dimers and the formation of weaker PEG–PMAA hydrogen bonds,

which diminish the electron-withdrawing environment of the carbonyl group³⁵. Peak-area ratio analysis further reveals that the fraction of strongly hydrogen-bonding carbonyl groups increases with increasing MAA content (**Fig. S3f**), underscoring the compositional dependence of hydrogen-bonding interactions within the gel network. Concomitantly, the C–O–C stretching of PEG (1000–1100 cm^{-1}) undergoes pronounced broadening and becomes increasingly asymmetric with rising MAA content, accompanied by the appearance of a shoulder at lower wavenumbers (**Fig. S4**)^{36,37}. This feature signifies the coexistence of free ether oxygen and hydrogen-bonding ether oxygen, reflecting hydrogen bond interaction between PEG ether oxygens and PMAA hydroxyl groups.

To elucidate the thermal evolution of this hierarchical hydrogen-bonding network at the molecular level, temperature-dependent FT-IR spectra of the PEM₂ gel were recorded from 30 to 150 °C. Focusing first on the O–H stretching region of PMAA (**Fig. 2d**), a gradual blue shift from 3500 to 3515 cm^{-1} indicates the progressive dissociation of hydrogen bond aggregates into free carboxyl groups. Two-dimensional correlation spectroscopy (2DCOS) provided higher resolution of these sequential events. While synchronous spectra (**Fig. 2e**) indicate cooperative spectral changes across the broad 2700–3600 cm^{-1} range, the asynchronous spectra (**Fig. 2f**) distinguish sequential thermal events: the spectral response of hydrogen bond O–H groups ($\sim 3297 \text{ cm}^{-1}$) precede that of the weakly associated or free O–H groups ($\sim 3596 \text{ cm}^{-1}$), corroborating the progressive thermal disruption of the network. Further insight is provided by the C=O stretching region (1690–1750 cm^{-1} , **Fig. S5**). The synchronous spectra exhibit positive cross-peaks between the band at $\sim 1700 \text{ cm}^{-1}$ (dimeric PMAA–PMAA hydrogen bonds) and the band at $\sim 1728\text{--}1730 \text{ cm}^{-1}$ (free C=O and PEG–PMAA hydrogen bonds), indicating concurrent intensity variations during heating. According to Noda’s rule, the positive synchronous and positive asynchronous correlations at (1700, 1728 cm^{-1}) reveals the specific reaction sequence: the destabilization of hydrogen-bonding PMAA–PMAA carbonyls initiates first, followed by the subsequent development of free or PEG–PMAA carbonyls.”

5. It remains uncertain whether the observed endothermic transitions are reversible. Please include cyclic DSC results to confirm the reversibility of hydrogen-bond dissociation and reassociation during heating–cooling cycles. This would provide stronger evidence for the proposed thermal buffering mechanism.

Response: We thank the reviewer for this valuable suggestion. In response, we have conducted cyclic

differential scanning calorimetry (DSC) measurements to explicitly evaluate the thermal reversibility of the endothermic transition, using the PE₆₀₀M₄ gel as a representative example (**Fig. S28**).

During the heating scan, a pronounced endothermic transition is observed, whereas no corresponding exothermic peak appears upon cooling. This result indicates that the thermal process is not fully reversible under the applied heating–cooling conditions. Such behavior is consistent with the intrinsic characteristics of the hierarchical hydrogen-bonding network in the gel. Specifically, the endothermic peak originates from the disruption of hierarchical hydrogen-bonding coupled with chain rearrangement. Once they are thermally dissociated, their reformation during cooling is kinetically hindered. Although hydrogen bonds can reform, the slow relaxation dynamics and chain-entanglement constraints prevent recovery of the original organization, resulting in the absence of a distinct exothermic peak. The updated results are now presented in **Fig. S28**, with the corresponding discussion revised on page 14 of the manuscript.

Fig. S28. Cyclic DSC curve of the PE₆₀₀M₄ gel between -30 to 150°C. The heating scan shows a clear endothermic peak, whereas no corresponding exothermic peak appears upon cooling.

Page 14: “This deconstructs the effective heat-buffering ability of the gels. A broad endothermic peak was also observed in the DSC curves of all PE_xM_y gels, attributed to the energy required for dissociation of the hierarchical hydrogen bonds (**Fig. 4b**). Integration of the DSC curves yielded the corresponding enthalpy values (**Fig. 4c**), which increased as the MAA content increased, consistent with the higher density of hydrogen bonds formed. To evaluate thermal reversibility, cyclic DSC measurement was conducted using the PE₆₀₀M₄ gel as an example (**Fig. S28**). The heating scan shows a clear endothermic peak, whereas no corresponding exothermic peak appears upon cooling. This behavior reflects the intrinsic nature of the hierarchical hydrogen-bonding network: the endothermic

peak originates from the disruption of hierarchical hydrogen-bonding coupled with chain rearrangement. Once they are thermally dissociated, their reformation during cooling is kinetically hindered. Although hydrogen bonds can reform, the slow relaxation dynamics and chain-entanglement constraints prevent recovery of the original organization, resulting in the absence of a distinct exothermic peak.”

6. The demonstration of the “M-glass” system is interesting, yet its long-term stability under realistic conditions (humidity, UV exposure, and thermal cycling) has not been discussed. Such data—or at least an informed discussion—are essential to evaluate its suitability for real-world glazing applications.

Response: We appreciate the reviewer’s important comment regarding the long-term stability of the M-glass system under realistic service conditions. In response, we have expanded the discussion and included representative environmental stability tests designed to simulate real-world environmental in practical applications.

Specifically, the M-glass was subjected to UV irradiation to simulate solar exposure, cyclic heating to mimic high-temperature condition, and water rinsing to emulate rainfall. Following these treatments, the M-glass consistently retained high optical transparency (**Fig. S37**), effective heat-absorption capability, and sound-insulation performance (**Fig. S38** and **Movie S7**), as well as robust impact resistance (**Fig. S39** and **Movie S8**). These results indicate that the M-glass exhibits good environmental tolerance and functional stability against representative UV, thermal, and humidity, supporting its suitability for real-world in practical applications. The updated results are now presented in **Fig. S37-39**, and **Movie S7, S8** with the corresponding discussion revised on **page 18** of the manuscript.

Page 18: *“To demonstrate the practical applicability of M-glass, we subjected it to representative environmental stressors: UV irradiation to simulate solar exposure, cyclic heating to mimic high-temperature condition, and water rinsing to emulate rainfall. Under all conditions, the M-glass consistently maintained high optical transparency (**Fig. S37**), effective heat-absorption, sound-insulation capability (**Fig. S38** and **Movie S7**), and robust impact resistance (**Fig. S39** and **Movie S8**), demonstrating excellent stability and functional reliability under realistic environmental scenarios.”*

Fig. S37. M-glass was irradiated under a UV lamp for 1 h, subjected to three heating-cooling cycles (heated at 40 °C for 2 h and cooled to room temperature), and rinsed under running water to mimic UV exposure, high-temperature condition, and rain condition, respectively. Thermal infrared images and photographs of all M-glass exposure to a xenon lamp. Scale bar: 2 cm.

Fig. S38. a-c. Experimental setup for sound measurement under UV, heat, and rain condition, respectively. Scale bar: 4 cm. d. Loudness variation of the blank group, original M-glass room, M-glass room under UV condition, heat condition, and rain condition.

Fig. S39. Photographs showing the M-glass under UV condition, heat condition, and rain condition after the impact of a 100 g steel ball dropped from a height of 30 cm. Scale bar: 2 cm.

7. For the falling-ball tests and glass-room drop tests, please add: replicates, error bars, and statistical treatment of impact force reductions.

Response: We thank the reviewer for this constructive suggestion. In response, we have incorporated the corresponding replicates, error bars, and statistical analyses for both the falling-ball and glass-room drop tests. Each measurement was performed in triplicate ($n = 3$), and the data are presented as mean \pm standard deviation (SD). The updated results are now presented in **Fig. 4e** and **Fig. S29**.

Fig. 4e. Summary of impact force reduction rates for steel balls of different weights dropped from a fixed height of 50 cm. Data are presented as the mean values \pm SD, $n = 3$ independent samples. Source data are provided as a Source Data file.

Fig. S29. Force profiles during falling-ball impacts and the reduction of impact forces by PE_xM_y gel coatings with steel balls of different weights. Data are presented as the mean values \pm SD, $n = 3$ independent samples. Source data are provided as a Source Data file.

8. Common glass interlayers include PVB and EVA. Please provide matched-thickness benchmarking against these commercial materials to contextualize performance. Clarify failure modes of the composite glass (glass cracking patterns, gel deformation) using high-speed images or post-impact observations.

Response: We thank the reviewer for the valuable suggestions. We acknowledge that PVB and EVA are widely used in glass interlayers; however, their performance and optical transparency are intrinsically coupled to specific processing routes.

Conventional PVB interlayers rely on prefabricated thermoplastic films followed by high-temperature, high-pressure autoclave processing. This process is essential to induce viscous flow, eliminate trapped air, ensure intimate interfacial contact, and suppress light scattering. Without such thermal–pressure treatment, PVB-laminated glass typically exhibits insufficient optical transmittance. EVA interlayers, while not requiring autoclave pressure, similarly depend on elevated-temperature lamination and irreversible thermal crosslinking of preformed films to achieve optical clarity and mechanical integrity. In the absence of adequate thermal curing, residual crystallinity and phase inhomogeneity in EVA lead to significant light scattering, preventing glass-grade transparency.

In contrast, the interlayer reported in this work is fabricated through direct injection of a low-viscosity liquid precursor between glass substrates, followed by *in situ* polymerization. This process enables complete wetting of the glass surfaces and intrinsic optical homogeneity from the outset, without the need for prefabricated films, high-temperature autoclaves, or post-lamination thermal curing.

Owing to this fundamentally different processing paradigm and optical formation mechanism, direct matched-thickness benchmarking against PVB or EVA films—which inherently require thermal lamination to become optically functional—is not pursued. Instead, we emphasize that the proposed gel-based interlayer represents a distinct and complementary strategy for laminated glass fabrication, prioritizing manufacturing simplicity, optical reliability, interfacial adaptability, and processing flexibility (**Table S2**).

Regarding failure modes, we have now included high-speed impact imaging (**Fig. S36** and **Movie S6**). Upon impact by a falling ball, the bare glass fractured immediately, whereas the glass protected by the PE₆₀₀M₄ gel remained fully intact, with no detectable damage to either component. So, the fracture of M-glass originates from stress concentration at its outer layer upon impact. In contrast, the

excellent damping capability of the gel effectively dissipates the impact energy, preventing its transmission to the inner glass layer and thereby maintaining the structural integrity of the entire M-glass. The updated results are now presented in **Fig. S36**, **Table S2**, and **Movie S6**, with the corresponding discussion revised on page 18-19 of the manuscript.

Supplementary Table 2. Comparison between the proposed in situ gel interlayer and conventional commercial glass interlayers.

Parameter	PVB	EVA	This work
Interlayer form	Polymer film	Polymer film	Liquid precursor
Construction method	Film stacking + autoclave	Film stacking + thermal curing	Direct injection + UV polymerization
Processing temperature	High (typically >120 °C)	Heating treatment (thermal crosslinking)	Room temperature
Processing pressure	High (autoclave required)	Low or none	None
Optical transparency	Achieved after thermal–pressure treatment	Achieved after thermal curing	Intrinsic from in situ polymerization
Interfacial adaptability	Limited (solid–solid contact)	Limited	Good (liquid wetting and in situ polymerization)
Manufacturing complexity	High	Moderate	Low
Equipment requirement	Autoclave	Heated laminator	Ultraviolet lamp

Fig. S36. Upon impact by a falling ball, the bare glass fractured immediately, whereas the glass protected by the PE₆₀₀M₄ gel remained fully intact, with no detectable damage to either component. So, the fracture of M-glass originates from stress concentration at its outer layer upon impact. In contrast, the excellent damping capability of the gel effectively dissipates the impact energy, preventing its transmission to the inner glass layer and thereby maintaining the structural integrity of the entire M-glass.

Page 18: “Finally, impact resistance was tested by dropping a 100 g ball from a height of 30 cm onto the glass rooms. As shown in **Fig. 5i** and **Movie S5**, N-glass shattered completely upon impact, whereas M-glass sustained only minor localized damage, confirming its protective function. *High-speed imaging shows the PE₆₀₀M₄ gel protects glass from impact by dissipating stress at the outer layer, preserving the glass’s structural integrity. (Fig. S36 and Movie S6).*”

Page 19: “Conventional PVB and EVA interlayers often requiring high-pressure autoclaves or irreversible thermal crosslinking to achieve optical transparency; without such treatment, trapped air, crystallinity, or phase inhomogeneity causes light scattering. Instead, we emphasize that the proposed gel-based interlayer represents a distinct and complementary strategy for laminated glass fabrication, prioritizing manufacturing simplicity, optical reliability, interfacial adaptability, and processing flexibility (**Table S2**).”

Reviewer #2:

This manuscript presents a PEG oligomer-based gel with functionalities in thermal regulation, sound attenuation, and mechanical protection. While the reported properties are interesting, the core concept — polymer oligomer solvent-based gels — is not novel. PEG-based “PEGgels” and related systems have been extensively studied in recent years, with detailed mechanistic, mechanical, and functional analyses already published (e.g., Nat. Commun. 2021, 12, 3610; Adv. Mater. 2022, 34, 2107791; Sci. Adv. 2025, 11, eadv5292). These works already have covered thermal stability, tunable mechanics, self-healing, energy dissipation, and protective applications. The authors highlight thermal regulation and glass interlayer application as new angles, but the overall novelty of this work requires further clarification based on the previous work. In other words, the use of PEG oligomers as solvent/structural components in gels is well-established. The manuscript does not convincingly demonstrate how this system advances beyond existing PEGgel literature, particularly in terms of mechanism or application scope. Therefore, I don't think the manuscript in current form is ready for publication in Nat. Commun..

Response: We thank the reviewer for pointing out these important prior studies. We agree that using polymer/oligomer fluids as the liquid phase of gels has been explored and has produced valuable insights and functionalities, including PEG-based gels and polymer-fluid-gel damping concepts. However, we emphasize that our manuscript advances beyond this literature in (i) molecular design strategy, (ii) mechanistic signature of dissipation, and (iii) application integration.

(i) Molecular design strategy (how hierarchy is generated).

The reported PEGgel work (Adv. Mater. 2022, 34, 2107791) uses a P(HEMA-co-AAc) matrix and highlights multivalent solvent-mediated interactions (notably abundant weak C–H···O hydrogen bonding) to enable toughness, self-healing, and printability. In contrast, our system is built on a PMAA backbone that intrinsically forms strong carboxylic-acid assemblies (dimers/aggregates) and then uses PEG-200 to introduce additional weaker/medium-strength H-bonding states. This creates a single-interaction-family hierarchy (hydrogen bonding) with a broadened strength distribution, rather than relying on comonomer-enabled network chemistry. In our manuscript, we further show that PMAA is essential for gelation in PEG, whereas replacing MAA with AAc prevents gelation (PEG/PAAc) (**Figure S2**), underscoring the distinct molecular requirement of the PMAA platform.

Fig. S2. Photographs illustrating the failure of gelation when MAA is replaced by AAc in the PEG-containing precursor solution. This because the alpha-methyl substituent increases hydrophobicity and stabilizes strong carboxyl to carboxyl associations^{1,2}. However, PAAc lacks the alpha-methyl group, prevents gelation in the PEA system, consistent with the loss of the strong-bond tier in the absence of robust self-association among carboxyl groups. Scale bar: 2 cm.

(ii) Mechanistic signature (what produces broadband damping).

The polymer-fluid-gel framework in Nat. Commun. 2021, 12, 3610 achieves controllable dissipation by tuning the reptation/relaxation time of confined polymer fluids, with relaxation time scaling with the molecular weight of the polymer fluid (reptation model) and producing frequency-dependent peaks that can be shifted by design. Our work targets a fundamentally different regime: we aim for nearly frequency-independent damping ($\tan \delta \approx 1$ across many decades via TTS) arising from a continuous spectrum of reversible hydrogen-bond association/dissociation events. We support this hierarchical H-bond mechanism using spectroscopic deconvolution/sequence analysis and MD results that quantify hydrogen-bond numbers, bond lengths, and binding energies for PMAA–PMAA versus PMAA–PEG interactions.

(iii) Application scope (integrated laminated-glass interlayer vs. single-function protection/damping).

Recent APFG work (Sci. Adv. 2025, 11, eadv5292) achieves broad-temperature damping and impact protection using PEG-based polymer-fluid gels combined with Fe^{3+} -diffusion surface ionic crosslinking to form an armor-core structure. Our manuscript instead demonstrates a transparent, adhesive, self-healable interlayer whose multifunctionality is intrinsically coupled to reversible hydrogen bonding: (1) viscoelastic damping, (2) impact energy absorption, and importantly (3) thermal buffering via endothermic H-bond dissociation, integrated into a laminated glass prototype. This “multifunction-in-one-interlayer” building-glazing scenario is not addressed in the cited studies, which primarily focus on toughness/printability or damping-protection under wide temperatures via armor-

core/ionic strengthening.

To make these distinctions explicit, we have revised the manuscript to position our work relative to PEGgel and polymer-fluid-gel literature, emphasizing that our key advance is oligomeric-solvent engineering of a PMAA-specific hierarchical hydrogen-bond landscape that enables simultaneous optical clarity + broadband damping + thermal buffering + interlayer adhesion in a single material system. The updated introduction revised on page 3 of the manuscript.

Page 3: *“Motivated by the inherent limitations of conventional small-molecule solvents, recent efforts have increasingly turned to polymeric liquids as alternative gel solvents. Liu et al.²¹ introduced poly (*n*-butyl acrylate) (PBA) fluid into polymer networks, achieving efficient energy dissipation over a broad frequency range. Wang et al.²²⁻²⁴ first established PEG as a polymeric solvent platform for gel systems, demonstrating that PEG-based gels can simultaneously exhibit high mechanical strength, self-healing capability, energy dissipation, and 3D printability. Although these studies highlight the versatility of polymeric solvents, their design targets and underlying mechanisms differ from those of the present work. In PEG gels based on P(HEMA-co-AAc)²³, enhanced toughness and self-healing primarily originate from solvent-enabled multivalent interactions and increased chain correlations or entanglements. In contrast, polymer-fluid-based damping gels²¹ achieve tunable dissipation by regulating the relaxation (reptation) dynamics of confined polymer liquids, leading to molecular-weight-dependent and frequency-shiftable dissipation peaks.*

*Here, we pursue a distinct molecular design strategy: we select PMAA, whose carboxylic groups intrinsically form strong associative assemblies²⁵ (bulk copolymerization of MAA yields glassy, brittle PMAA, **Fig. 1a**), and employ PEG-200 as an oligomeric solvent that introduces additional weaker/medium-strength hydrogen-bonding states, thereby broadening the bond-strength distribution within a single interaction family (hydrogen bonding). First, owing to its oligomeric nature, PEG has far greater contour length than small-molecule solvents and more effectively separate adjacent PMAA chains, thereby increasing interchain spacing and free volume^{26,27}. Second, the ether oxygen on PEG bear lone pairs and serve as hydrogen-bonding acceptors, forming multiple bidentate (medium-strength) and monodentate (weak) interactions with the carboxylic groups of PMAA²⁸. In parallel, a fraction of PMAA carboxyl groups associate with each other from cyclic dimers to linear aggregates^{25,29}, supplying the strong-bond tier of the hierarchy. (**Fig. 1b**). Notably, both PEG and PMAA are indispensable for building this hierarchical hydrogen-bonding network. Substituting PEG*

with H₂O results in a glassy, brittle PMAA material rather than a gel (Fig. S1), while replacing PMAA with polyacrylic acid (PAAc) suppresses gelation in the PEG/PAAc (PEA) system (Fig. S2). This architecture achieves finely balance elasticity and viscosity^{30,31}, enabling a gel-like response across an ultra-wide frequency range (Fig. 1c). The elastic component provides high mechanical strength, while the viscous component delivers high damping, with a loss factor ($\tan \delta$) approaching 1. This combination confers both noise reduction and impact energy absorption. Simultaneously, the thermally responsive interactions of the PEG/PMAA system enhance thermal regulation. To demonstrate practical utility, we integrated the gel into a laminated-glass prototype and systematically evaluated its multifunctionality (Fig. 1d). The resulting composite exhibits combination of optical clarity, thermal buffering, acoustic damping, and mechanical robustness. Rather than optimizing isolated properties, this system achieves integrated performance, opening avenues for functional materials in architectural glazing, protective interfaces, and energy-responsive systems.”

1. The comparison to starfish stiffness modulation is misleading. Starfish employ complex, reversible, biologically regulated systems (blood pressure, peptides); the PEM gel simply softens via PEG plasticization — a fundamentally different and far less sophisticated mechanism.

Response: We thank the reviewer for this insightful comment and fully agree with the concern raised. Stiffness regulation in starfish relies on complex, reversible, and biologically regulated mechanisms that are not directly comparable to the physicochemical behavior of our synthetic system. Accordingly, all biomimetic references to starfish have been removed. The revised manuscript now focuses exclusively on the role of PEG oligomers as molecular bridges that separate polymer chains and construct a hierarchical hydrogen-bonding network, thereby eliminating any potentially misleading biological analogy.

2. Page 12, Line 261. The explanations about elasticity and viscosity (consistent damping factor) across broad temperatures lack scientific theories or experiment support, authors can refer to some previous work (e.g. Nat. Commun. 2021, 12(1), 3610).

Response: We thank the reviewer for this insightful comment. We agree that a clearer theoretical framework is necessary to rationalize the observed quasi-constant damping behavior over broad frequency and temperature ranges.

Inspired by the reviewer’s suggestion, we now explicitly interpret our results within the framework of hierarchical relaxation mechanisms, analogous in principle to polymer-fluid-gel systems reported previously (e.g., Nat. Commun. 2021, 12, 3610), while emphasizing the fundamental differences in physical origin.

In our PEG/PMAA system, the balanced viscoelastic response does not arise from confined polymer fluid reptation, but from a hierarchical hydrogen-bonding network that introduces multiple, overlapping relaxation modes. Specifically, strong PMAA–PMAA hydrogen bonds provide long-lived elastic constraints, medium-strength PMAA–PEG interactions contribute intermediate relaxation, and weak hydrogen bonds together with chain entanglements enable fast energy dissipation. The coexistence and continuous exchange among these relaxation processes lead to comparable storage and loss moduli across a wide frequency window, resulting in a loss factor fluctuating around unity.

This interpretation is experimentally supported by:

(i) Time–temperature superposition master curves showing $G' \approx G''$ over nearly eight orders of magnitude in frequency.

(ii) Two distinct apparent activation energies extracted from rheological analysis, corresponding to bond-dissociation-dominated and entanglement-dominated relaxation regimes.

(iii) Temperature-dependent FTIR and 2D correlation spectroscopy revealing stepwise dissociation of hydrogen bonds with different strengths rather than a single glass-transition-controlled process.

We have revised the manuscript to clarify this theoretical connection and to more explicitly link our experimental observations to established viscoelastic relaxation principles (Pages S11–S12 and Fig. S19).

Fig. S19. Internal network state of PE_xM_y gels under different shear frequency.

Page S11: “**Fig. S19** schematically illustrates the physical origin of the quasi-frequency- and temperature-independent viscoelastic response of PEM_y gels within the framework of time-temperature equivalence and relaxation-spectrum theory.

In viscoelastic polymer networks, the macroscopic balance between elasticity and viscosity is governed by the superposition of multiple relaxation modes with distinct characteristic times. When these relaxation modes are narrowly distributed, energy dissipation is localized near the glass transition, leading to a sharp peak in $\tan \delta$. In contrast, a broad and overlapping relaxation spectrum can give rise to a nearly constant loss factor over extended frequency and temperature ranges.

In PEM_y gels, such a broad relaxation spectrum originates from the hierarchical hydrogen-bonding network combined with chain entanglements. At low frequencies (or equivalently high temperatures, **Fig. S19a**), weak and medium-strength hydrogen bonds dissociate first, increasing chain mobility and releasing long network strands. Although the effective density of elastic constraints decreases due to bond dissociation, the liberated chains simultaneously form additional topological entanglements. This compensatory mechanism preserves the balance between energy storage and dissipation, preventing a transition to purely viscous flow. At intermediate frequencies (**Fig. S19b**), partial rupture of weak hydrogen bonds occurs while strong PMAA–PMAA hydrogen bonds and chemical crosslinks remain intact. In this regime, hydrogen-bond dissociation and entanglement formation coexist dynamically, establishing a steady-state viscoelastic response in which the relaxation times of different modes overlap. As a result, both G' and G'' remain comparable in magnitude. At high frequencies (or low temperatures, **Fig. S19c**), molecular motion is increasingly constrained, suppressing chain reptation and entanglement relaxation. However, the dominance of strong hydrogen bonds and physical crosslinking effectively “freezes” the network on the experimental timescale, sustaining elastic energy storage. Consequently, although individual relaxation processes are restricted, the collective contribution of long-lived elastic modes compensates for the reduced viscous dissipation.

Overall, the hierarchical hydrogen-bond architecture generates a continuous distribution of relaxation times spanning bond dissociation, chain sliding, and entanglement relaxation. The superposition of these modes underlies the observed $G' \approx G''$ and $\tan \delta \approx 1$ behavior across an ultra-wide frequency and temperature window, analogous in principle to multi-relaxation damping systems, yet achieved here through dynamic supramolecular interactions rather than confined polymer fluids.”

3. Figure 3h. Typically, to evaluate the damping capacity of materials at varied tensile strains, the tensile tests are performed at separated strain cycles instead of consecutive cycles.

Response: We thank the reviewer for this helpful comment. In accordance with the reviewer’s recommendation, the damping capacities previously derived from consecutive cyclic tensile tests have been removed. Independent single-cycle tensile tests were instead conducted at each fixed strain, and the damping capacity was recalculated for each individual strain.

In addition, large-amplitude oscillatory shear (LAOS) measurements were conducted to further evaluate the energy dissipation behavior of the PEM_y gels (Fig. S24). All samples exhibit stable, elliptical stress–strain Lissajous loops over 100 cycles, indicating a linear viscoelastic response under the applied conditions. The enclosed loop area, corresponding to the dissipated energy per cycle (ΔW), shows only a slight decrease during repeated cycling, suggesting that the reversible hydrogen bonds undergo repeated dissociation and reformation without causing permanent structural damage. Moreover, ΔW systematically increases with increasing MAA content, demonstrating that a higher density of hydrogen-bonding interactions leads to enhanced energy dissipation. The updated results are now presented in Fig. 3g, Fig. 3h, Fig. S23, and Fig. S24 with the corresponding discussion revised on page 13 of the manuscript.

Fig. 3. Mechanical properties and viscoelastic behavior of PEM_y gels. **g.** Tensile load-unload cycle curves of the PEM₂ gel at different fixed strain. **h.** Summary of energy dissipation values and damping capacity of the PEM₂ gel with different tensile strains. **i.** Summary of energy dissipation values and damping capacity of all PEM_y gel at strain of 1mm/mm. Data are presented as the mean values \pm SD, $n = 3$ independent samples. Source data are provided as a Source Data file.

Fig. S23. a-c. Tensile load-unload cycle curves of PEM_y gel samples at different fixed strain. d-e. Summarized energy dissipation values and damping capacity of PEM_y gel samples with different tensile load-unload cycle. Data are presented as the mean values \pm SD, n = 3 independent samples. Source data are provided as a Source Data file.

Fig. S24. Large amplitude oscillatory shear (LAOS) tests of the PEM_y gels. The hysteresis loops are elliptical, indicating all gel are located in the linear viscoelastic region with 10% strain^{4,5}. The hysteresis loop area represents mechanical energy converted to heat during each cycle and the area decreases only slightly after 100 cycles.

Page 13: “As an intuitive manifestation of the damping capability, energy dissipation serves as a key criterion for evaluating the damping performance of materials. In tensile loading–unloading cycles, the hysteresis loops enlarge with strain (Fig. 3g and S23a-c). The damping capacity, defined as the

ratio of dissipated energy to total strain energy, decreases with increasing strain (**Fig. 3h** and **S23d-e**). There are two factors contributing to this result. First, at small strain, the polymer chains remain coiled, enabling substantial internal friction and interchain sliding, which leads to pronounced energy dissipation and high damping capacity. With increasing strain, chains gradually elongate and orient, reducing available sliding space and thereby lowering frictional dissipation. Second, at small strain, the dynamic and reversible crosslinks within the gel can continuously break and reform, contributing significantly to energy dissipation. With increasing strain, many of these weak interactions are rapidly ruptured and cannot readily reform, thereby diminishing the dissipation capacity. Meanwhile, the network is increasingly supported by the polymer backbone and strong crosslinks, causing the gel network to exhibit more "elasticity". Notably, we observed at same strain, the damping capacity increases with MAA content (**Fig. 3i**), underscoring the key role of hydrogen bond density in enhancing energy dissipation.

Consistent with tensile observations, large-amplitude oscillatory shear (LAOS) tests further demonstrate the dissipation capability of the PEM_y gels (**Fig. S24**). All gels exhibit stable elliptical Lissajous loops over 100 cycles, evidencing stable linear viscoelastic behavior⁴⁹. The loop area, corresponding to the dissipated energy per cycle (ΔW), decreases only slightly during cycling, confirming that the reversible hydrogen bonds repeatedly break and reform without structural damage. Moreover, ΔW increases with MAA content, demonstrating higher density of hydrogen-bonding enhances energy dissipation.”

4. The explanations on the spectroscopy should be more comprehensive. How is the formation of H-bonds between PEG and MAA resulting in the blueshift? What about the corresponding shift for PEG.

Response: We thank the reviewer for this constructive comment. We have expanded the FT-IR spectroscopy discussion to clarify the origin of the observed peak shifts and to provide a more comprehensive interpretation of the hydrogen-bonding interactions between PEG and MAA. Furthermore, we have incorporated systematic FT-IR analysis combined with temperature-dependent two-dimensional infrared (2D-IR) spectroscopy to directly elucidate the hierarchical hydrogen-bonding network in the gel.

The blue shift observed for the C=O stretching vibration of MAA originates from the transformation of hydrogen-bonding environments upon PEG incorporation. In neat MAA, C=O

predominantly form cyclic, symmetrical PMAA–PMAA dimers through strong O–H···O=C···H–O hydrogen bonds. These strong dimeric interactions significantly withdraw electron density from the C=O, weaken the C=O bond, and consequently shift the stretching vibration to lower wavenumbers (typically around $\sim 1700\text{ cm}^{-1}$, **Fig. S3a**).

Upon the introduction of PEG, part of PMAA–PMAA dimers is disrupted and replaced by PEG–PMAA hydrogen bonds, in which the ether oxygen of PEG serves as the hydrogen-bond acceptor and the MAA hydroxyl group as the donor. Compared with the highly ordered PMAA–PMAA dimers, PEG–PMAA hydrogen bonds are weaker and less symmetric. So, the electron-withdrawing effect on the carbonyl group is reduced, leading to an increase in C=O bond strength and a corresponding shift of the stretching vibration toward higher wavenumbers. This manifests as the emergence of an additional blue-shifted peak near $\sim 1728\text{ cm}^{-1}$ (**Fig. S3b-e**). Furthermore, quantitative peak-area ratio analysis demonstrates that the relative fraction of strongly hydrogen-bonded carbonyl groups increases monotonically with increasing MAA content (**Fig. S3f**), providing direct evidence for the composition-dependent hierarchy of hydrogen-bond interactions within the gel.

Correspondingly, changes are also observed in the PEG vibrational signatures. The C–O–C stretching band of PEG ($1000\text{--}1100\text{ cm}^{-1}$) undergoes pronounced broadening and becomes increasingly asymmetric with increasing MAA content, accompanied by the appearance of a shoulder at lower wavenumbers (**Fig. S4**). Rather than exhibiting a distinct peak shift, this behavior reflects the coexistence of free and hydrogen-bonded ether oxygen environments. The shoulder at lower wavenumbers is attributed to PEG ether oxygens engaged in hydrogen bonding with PMAA hydroxyl groups, while the main band corresponds to non-interacting or weakly interacting ether oxygens. This spectral evolution signifies the coexistence of free ether oxygens and hydrogen-bonded ether oxygens, confirming the formation of PEG–PMAA hydrogen bonds and further supporting the heterogeneous hydrogen-bonding environment.

Then, the thermal evolution of the hierarchical hydrogen-bond network was directly probed by temperature-dependent FT-IR and 2D correlation infrared spectroscopy. FT-IR spectra of the PEM₂ gel were collected during heating from 30 to 150 °C. In the O–H stretching region of PMAA (**Fig. 2d**), a gradual blue shift from ~ 3500 to $\sim 3515\text{ cm}^{-1}$ is observed, indicating progressive dissociation of hydrogen-bonding aggregates into weaker associated or free carboxyl groups.

To resolve the sequential nature of these thermal events, two-dimensional correlation spectroscopy

(2DCOS) was employed. The synchronous spectrum (**Fig. 2e**) reveals cooperative spectral changes across the broad 2700–3600 cm^{-1} region, confirming the collective response of hydrogen-bonded species to thermal perturbation. More critically, the asynchronous spectrum (**Fig. 2f**) distinguishes the order of molecular events: the response of strongly hydrogen-bonded O–H groups ($\sim 3297 \text{ cm}^{-1}$) precede that of weakly associated or free O–H groups ($\sim 3596 \text{ cm}^{-1}$), directly evidencing stepwise thermal dissociation of hydrogen bonds with different strengths.

Additional mechanistic insight is obtained from the carbonyl stretching region (1690–1750 cm^{-1} , **Fig. S5**). The synchronous 2D-IR spectrum shows positive cross-peaks between the $\sim 1700 \text{ cm}^{-1}$ band (dimeric PMAA–PMAA hydrogen bonds) and the $\sim 1728\text{--}1730 \text{ cm}^{-1}$ band (free or PEG–PMAA-associated carbonyls), indicating correlated intensity changes upon heating. According to Noda’s rule, the presence of both positive synchronous and positive asynchronous cross-peaks at (1700, 1728 cm^{-1}) establishes a well-defined sequence: the disruption of strong PMAA–PMAA hydrogen bonds occurs first, followed by the emergence of weaker or free carbonyl species. This spectroscopic evidence unambiguously confirms the hierarchical and discontinuous nature of the hydrogen-bond network.

Collectively, these results provide direct experimental validation of the proposed multi-tiered hydrogen-bonding mechanism, thereby substantially strengthening the mechanistic foundation of the manuscript. The updated results are now presented in **Fig. 2d-f**, **Fig. S3-S5**, with the corresponding discussion revised on **page 7** of the manuscript.

Fig. 2. Characterizations and simulations revealing hierarchical hydrogen-bonding in PEM_y gels. d. Temperature-dependent FTIR spectra of PEM₂ gel upon heating from 30 to 150 °C (10 °C intervals). **e, f. Synchronous and asynchronous 2DCOS spectra of the PEM₂ gel.** Red regions indicate positive intensities, while blue regions represent negative ones.

Fig. S3. Peak fitting analysis of the carbonyl stretching vibration [$\nu(\text{C}=\text{O})$] at 1700 cm⁻¹. **a.** Pure MAA. **b-e.** PEM_y gel, $y=1.8, 2, 2.2,$ and 2.4 . **f.** Corresponding area fractions of the fitted carbonyl peak.

Figure S4. Enlarged FT-IR spectra in the C-O-C stretching region (1000–1100 cm⁻¹) of neat PEG and PEM_y gels.

Fig. S5. Synchronous and asynchronous 2DCOS spectra of the PEM₂ gel. Red regions indicate positive intensities, while blue regions represent negative ones.

Page 7: “Fourier transform infrared (FT-IR) spectroscopy further elucidate the chemical composition and interaction mechanism within the PEM_y gels (**Fig. 2c**). For neat PEG, absorption peaks at 2876 cm⁻¹ and 2930 cm⁻¹ correspond to symmetrical and asymmetric stretching of -CH₂ groups, while the peak at 1059 cm⁻¹ is assigned to C-O-C stretching^{33,34}. For neat MAA, the characteristic C=O stretching vibration appears near 1700 cm⁻¹. These characteristic peaks are present in all PEM_y gel samples, confirming the successful incorporation of both components.

To clarify the hierarchical nature of the hydrogen-bonding network, peak deconvolution was performed on the C=O vibration FT-IR spectra. Neat MAA exhibits a prominent carbonyl peak at 1700 cm⁻¹ (**Fig. S3a**), characteristic of strong dimeric hydrogen bonds. Upon PEG incorporation, an additional blue-shifted peak emerges near 1728 cm⁻¹ (**Fig. S3b-e**). This shift arises from the disruption of partial strong PMAA-PMAA dimers and the formation of weaker PEG-PMAA hydrogen bonds, which diminish the electron-withdrawing environment of the carbonyl group³⁵. Peak-area ratio analysis further reveals that the fraction of strongly hydrogen-bonding carbonyl groups increases with increasing MAA content (**Fig. S3f**), underscoring the compositional dependence of hydrogen-bonding interactions within the gel network. Concomitantly, the C-O-C stretching of PEG (1000–1100 cm⁻¹) undergoes pronounced broadening and becomes increasingly asymmetric with rising MAA content, accompanied by the appearance of a shoulder at lower wavenumbers (**Fig. S4**)^{36,37}. This feature signifies the coexistence of free ether oxygen and hydrogen-bonding ether oxygen, reflecting hydrogen bond interaction between PEG ether oxygens and PMAA hydroxyl groups.

To elucidate the thermal evolution of this hierarchical hydrogen-bonding network at the molecular level, temperature-dependent FT-IR spectra of the PEM₂ gel were recorded from 30 to 150 °C. Focusing first on the O-H stretching region of PMAA (**Fig. 2d**), a gradual blue shift from 3500 to 3515 cm⁻¹ indicates the progressive dissociation of hydrogen bond aggregates into free carboxyl groups. Two-dimensional correlation spectroscopy (2DCOS) provided higher resolution of these sequential events. While synchronous spectra (**Fig. 2e**) indicate cooperative spectral changes across the broad 2700–3600 cm⁻¹ range, the asynchronous spectra (**Fig. 2f**) distinguish sequential thermal events: the spectral response of hydrogen bond O-H groups (~3297 cm⁻¹) precede that of the weakly associated or free O-H groups (~3596 cm⁻¹), corroborating the progressive thermal disruption of the network.

Further insight is provided by the C=O stretching region (1690–1750 cm⁻¹, **Fig. S5**). The synchronous spectra exhibit positive cross-peaks between the band at ~1700 cm⁻¹ (dimeric PMAA-

PMAA hydrogen bonds) and the band at ~1728–1730 cm⁻¹ (free C=O and PEG–PMAA hydrogen bonds), indicating concurrent intensity variations during heating. According to Noda's rule, the positive synchronous and positive asynchronous correlations at (1700, 1728 cm⁻¹) reveals the specific reaction sequence: the destabilization of hydrogen-bonding PMAA–PMAA carbonyls initiates first, followed by the subsequent development of free or PEG–PMAA carbonyls.”

5. The manuscript introduces asymmetric adhesion without explaining its functional purpose, to me it appears to offer only drawbacks.

Response: We thank the reviewer for this thoughtful comment. We clarify that the asymmetric adhesion observed in the preliminary samples is not an intended functional feature, but rather an artifact arising from the specific polymerization conditions employed during early-stage characterization. In these samples, one surface of the gel was in contact with PTFE, while the opposite surface was exposed to air, resulting in distinct interfacial environments and consequently asymmetric surface adhesion.

Importantly, this asymmetric adhesion is not relevant to the intended application of the material. For practical use as a glass interlayer, the precursor solution is injected directly between two glass substrates and polymerized in situ. Under these conditions, both interfaces are formed under identical environments, leading to symmetric and significantly enhanced adhesion. As demonstrated in **Movie S3**, the in situ–polymerized gel adheres so strongly on the glass surfaces that cannot be separated without damage. A circular gel (diameter = 2 cm, thickness = 1 mm) was polymerized in situ between two glass substrates. Upon application of a 500 g load, no adhesive failure was observed (**Fig. 5a**). Remarkably, the laminated structure was capable of lifting a 2 kg weight. The adhesive joint remained fully intact, while fracture occurred in the glass substrate (**Movie S3**).

To enable quantitative evaluation of adhesion properties, we polymerized the precursor solution between PET release films, ensuring identical interfacial conditions on both sides of the gel. Adhesive performance was quantitatively evaluated using 90° peel and lap-shear tests. Notably, the PE₆₀₀M₄ gel exhibited strong adhesion across various substrates (**Fig. S34**), achieving an adhesion strength and interfacial toughness on glass as high as 332 kPa and 676 J m⁻², respectively (**Fig. 5b**). In addition, the gel demonstrated strong yet compliant adhesion to human skin (**Fig. S35**).

Furthermore, cyclic lap-shear tests conducted within ±5% strain revealed stable dynamic stress

responses, with the gel sustaining at least 1000 cycles of shear deformation without interfacial failure (Fig. 5c), confirming the robustness and durability of the gel–glass interface. The updated results are now presented in Fig. 5a-c, Fig. S33 and Movie S3, with the corresponding discussion revised on page 17 of the manuscript.

Fig. 5. Multifunctional laminated glass composite enabled by oligomeric-solvent-containing PMAA gels. a. Photographs showing the interlayer region (circled) remaining intact during lifting of a 50 g weight. Scale bar: 2 cm. **b.** Summary of adhesion strength and interfacial toughness of PE₆₀₀M₄ gel samples on different substrates. **c.** Lap-shear stress curve during the repeated loading-unloading cycles.

Fig. S33. Adhesion performance of the PE₆₀₀M₄ gel. a. Photographs demonstrating a square PE₆₀₀M₄ gel (1×1 cm²) adhered firmly to various substrates. On glass, polyethylene terephthalate (PET), and copper, the adhesion was sufficient to readily lift a 200 g weight. Although adhesion on rubber was slightly weaker, it remained effective. Scale bar: 4 cm. **b.** 90° peeling curves of the PE₆₀₀M₄ gel on various substrates. **c.** Shear adhesion stress-displacement curves of the PE₆₀₀M₄ gel on various substrates.

Page 17: “As glass interlayers, strong glass–interlayer adhesion is essential for the long-term integrity and multifunctional stability of safety glass. Owing to the abundance of hydroxyl groups, the PE_xM_y gel exhibits robust adhesion. When polymerized in situ on glass, the resulting adhesion is too strong to separate them. To demonstrate this strong adhesion, a circular gel (diameter = 2 cm, thickness = 1 mm) was polymerized in situ between two glass substrates. Upon application of a 500 g load, no adhesive failure was observed (Fig. 5a and Movie S3). Remarkably, the laminated structure was

capable of lifting a 2 kg weight. The adhesive joint remained fully intact, while fracture occurred in the glass substrate. To avoid asymmetric adhesion arising from differences in polymerization environments at the two interfaces⁵¹, PE₆₀₀M₄ gels were polymerized between PET release film. Adhesive performance was quantitatively evaluated using 90° peel and lap-shear tests, the PE₆₀₀M₄ gel demonstrated strong adhesion on various substrates (Fig. S33), achieving an adhesion strength and interfacial toughness on glass as high as 332 kPa and 676 J m⁻² (Fig. 5b), respectively. In addition, the gel also exhibited strong yet compliant adhesion to human skin (Fig. S34). Cyclically shearing of the PE₆₀₀M₄ gel within ±5% strain generated stable dynamic stress responses, the gel could sustain at least 1000 cycles of lap-shearing deformations (Fig. 5c).”

6. Missing figure captions and wrongly assigned figure numbers for figures S8 and after.

Response: We sincerely thank the reviewer for the careful examination of the manuscript. We apologize for the missing figure captions and the incorrect figure numbering starting from Fig. S8. All captions have now been restored, and the figure numbering has been carefully checked and corrected throughout the revised Supplementary Information.

Reviewer #3:

The manuscript by Li et al. reports a polymeric material composed of poly(methacrylic acid) (PMAA) swollen in poly(ethylene glycol) (PEG), forming a gel-like network that exhibits properties including high energy dissipation, acoustic damping, self-healing, adhesiveness, and impact resistance. These properties arise from the interaction between PEG macromolecules and the PMAA network. The material and its performance are well characterized, and the authors demonstrate several proof-of-concept applications. Overall, the experimental work is solid and the results are of potential interest. However, the presentation of prior work is incomplete and somewhat misleading. The concept of using PEG as a macromolecular solvent was first introduced by Wang Z. et al. (Adv. Mater. 2022, “Tough, Transparent, 3D-Printable, and Self-Healing Poly(ethylene glycol)-Gel (PEGgel)”), where the fundamental principles, transparency, self-healing, and exceptional mechanical properties were already demonstrated. Subsequent studies, including Wang M. et al., Nature 2024, have further explored applications and properties of PEG–polymer hybrid materials.

The authors should therefore revise the Abstract and Introduction to properly acknowledge this prior

work and avoid implying that the present study introduces the concept for the first time (e.g., phrases such as “our oligomeric-solvent engineering strategy” or “we introduce oligomeric PEG into a PMAA network to construct a gel system that addresses key limitations of traditional solvent-based gels” are misleading).

With appropriate correction of the contextual framing, this manuscript would represent a valuable contribution to the field highlighting the advantages and further applications of an already established concept.

Response: We sincerely thank the reviewer for the careful evaluation of our manuscript and for pointing out the important prior work on PEG-based macromolecular solvent systems. We fully agree that the concept of using PEG as a macromolecular solvent was first introduced by Wang Z. et al. (Adv. Mater. 2022, 34, 2107791).

We apologize for the inappropriate wording in the original Abstract and Introduction. Following the reviewer’s suggestion, we have revised the Abstract and Introduction to explicitly cite these pioneering studies and to clearly position our work as building upon and extending the established macromolecular solvent concept, rather than introducing it.

We would like to clarify that the novelty of the present work does not lie in introducing PEG as a macromolecular solvent, but rather in the following three aspects:

(i) Molecular design strategy (how hierarchy is generated).

The reported PEGgel work (Adv. Mater. 2022, 34, 2107791) uses a P(HEMA-co-AAc) matrix and highlights multivalent solvent-mediated interactions (notably abundant weak C–H···O hydrogen bonding) to enable toughness, self-healing, and printability. In contrast, our system is built on a PMAA backbone that intrinsically forms strong carboxylic-acid assemblies (dimers/aggregates) and then uses PEG-200 to introduce additional weaker/medium-strength H-bonding states. This creates a single-interaction-family hierarchy (hydrogen bonding) with a broadened strength distribution, rather than relying on comonomer-enabled network chemistry. In our manuscript, we further show that PMAA is essential for gelation in PEG, whereas replacing MAA with AAac prevents gelation (PEG/PAAc) (**Figure S2**), underscoring the distinct molecular requirement of the PMAA platform.

Fig. S2. Photographs illustrating the failure of gelation when MAA is replaced by AAc in the PEG-containing precursor solution. This because the alpha-methyl substituent increases hydrophobicity and stabilizes strong carboxyl to carboxyl associations^{1,2}. However, PAAc lacks the alpha-methyl group, prevents gelation in the PEA system, consistent with the loss of the strong-bond tier in the absence of robust self-association among carboxyl groups. Scale bar: 2 cm.

(ii) Mechanistic signature (what produces broadband damping).

Our work targets a fundamentally different regime: we aim for nearly frequency-independent damping ($\tan \delta \approx 1$ across many decades via TTS) arising from a continuous spectrum of reversible hydrogen-bond association/dissociation events. We support this hierarchical H-bond mechanism using spectroscopic deconvolution/sequence analysis and MD results that quantify hydrogen-bond numbers, bond lengths, and binding energies for PMAA–PMAA versus PMAA–PEG interactions.

(iii) Application scope (integrated laminated-glass interlayer vs. single-function protection/damping).

Our manuscript demonstrates a transparent, adhesive, self-healable interlayer whose multifunctionality is intrinsically coupled to reversible hydrogen bonding: (1) viscoelastic damping, (2) impact energy absorption, and importantly (3) thermal buffering via endothermic H-bond dissociation, integrated into a laminated glass prototype. This integrated functionality distinguishes the present system from prior PEG-based gels that primarily target single-function mechanical protection or damping applications.

To make these distinctions explicit, we have revised the Abstract and Introduction. We now emphasize that our key advance is oligomeric-solvent engineering of a PMAA-specific hierarchical hydrogen-bond landscape that enables simultaneous optical clarity + broadband damping + thermal buffering + interlayer adhesion in a single material system. The updated introduction revised on page 3 of the manuscript.

Page 3: “*Motivated by the inherent limitations of conventional small-molecule solvents, recent efforts*

have increasingly turned to polymeric liquids as alternative gel solvents. Liu et al.²¹ introduced poly (*n*-butyl acrylate) (PBA) fluid into polymer networks, achieving efficient energy dissipation over a broad frequency range. Wang et al.²²⁻²⁴ first established PEG as a polymeric solvent platform for gel systems, demonstrating that PEG-based gels can simultaneously exhibit high mechanical strength, self-healing capability, energy dissipation, and 3D printability. Although these studies highlight the versatility of polymeric solvents, their design targets and underlying mechanisms differ from those of the present work. In PEG gels based on P(HEMA-co-AAc)²³, enhanced toughness and self-healing primarily originate from solvent-enabled multivalent interactions and increased chain correlations or entanglements. In contrast, polymer-fluid-based damping gels²¹ achieve tunable dissipation by regulating the relaxation (reptation) dynamics of confined polymer liquids, leading to molecular-weight-dependent and frequency-shiftable dissipation peaks.

Here, we pursue a distinct molecular design strategy: we select PMAA, whose carboxylic groups intrinsically form strong associative assemblies²⁵ (bulk copolymerization of MAA yields glassy, brittle PMAA, **Fig. 1a**), and employ PEG-200 as an oligomeric solvent that introduces additional weaker/medium-strength hydrogen-bonding states, thereby broadening the bond-strength distribution within a single interaction family (hydrogen bonding). First, owing to its oligomeric nature, PEG has far greater contour length than small-molecule solvents and more effectively separate adjacent PMAA chains, thereby increasing interchain spacing and free volume^{26,27}. Second, the ether oxygen on PEG bear lone pairs and serve as hydrogen-bonding acceptors, forming multiple bidentate (medium-strength) and monodentate (weak) interactions with the carboxylic groups of PMAA²⁸. In parallel, a fraction of PMAA carboxyl groups associate with each other from cyclic dimers to linear aggregates^{25,29}, supplying the strong-bond tier of the hierarchy. (**Fig. 1b**). Notably, both PEG and PMAA are indispensable for building this hierarchical hydrogen-bonding network. Substituting PEG with H₂O results in a glassy, brittle PMAA material rather than a gel (**Fig. S1**), while replacing PMAA with polyacrylic acid (PAAc) suppresses gelation in the PEG/PAAc (PEA) system (**Fig. S2**). This architecture achieves finely balance elasticity and viscosity^{30,31}, enabling a gel-like response across an ultra-wide frequency range (**Fig. 1c**). The elastic component provides high mechanical strength, while the viscous component delivers high damping, with a loss factor ($\tan \delta$) approaching 1. This combination confers both noise reduction and impact energy absorption. Simultaneously, the thermally responsive interactions of the PEG/PMAA system enhance thermal regulation. To demonstrate

practical utility, we integrated the gel into a laminated-glass prototype and systematically evaluated its multifunctionality (Fig. 1d). The resulting composite exhibits combination of optical clarity, thermal buffering, acoustic damping, and mechanical robustness. Rather than optimizing isolated properties, this system achieves integrated performance, opening avenues for functional materials in architectural glazing, protective interfaces, and energy-responsive systems.”

Dear Reviewers,

Thank you very much for reviewing our manuscript entitled “*Oligomeric-Solvent Engineering of Hierarchical Hydrogen-Bond Networks for Multifunctional Glass Interlayers*” (Manuscript ID: NCOMMS-25-73988A). We sincerely appreciate your time and the constructive comments provided.

We are particularly grateful for your careful assessment and constructive feedback throughout the review process, which have been instrumental in improving the rigor, clarity, and overall quality of the manuscript. Your insightful comments have significantly strengthened the presentation and impact of our study.

Thank you again for your valuable contributions and support.

Sincerely,

Rong Ran

Response to the reviewers' comments

Reviewer #1: The revised manuscript is suitable for publication.

Response: We sincerely thank reviewer for the careful re-evaluation of our revised manuscript and for the positive assessment. We greatly appreciate your constructive comments throughout the review process, which have significantly improved the clarity and quality of the work.

Reviewer #2: The authors have thoroughly addressed the reviewers' critiques by significantly enhancing the mechanistic evidence through additional FT-IR, 2D-COS, and MD simulation data, which explicitly validate the hierarchical hydrogen-bonding network. They have also clearly acknowledged relevant prior work and differentiated their work from prior PEG-gel studies by emphasizing the single-interaction-family design and broadened damping performance, supported by new environmental stability tests and quantitative impact analyses. The revisions have strengthened the manuscript's novelty, data completeness, and practical relevance, making it suitable for publication.

Response: We sincerely thank the reviewer for the thorough evaluation and highly encouraging comments. We greatly appreciate your recognition of the additional mechanistic analyses, including the FT-IR, 2D-COS, and MD simulation results, as well as your acknowledgment of our efforts to

clarify the distinction from prior PEG-gel studies and to strengthen the environmental stability and quantitative impact assessments. Your constructive feedback has been invaluable in improving the rigor, clarity, and overall quality of the manuscript.